



# Data-driven estimates of evapotranspiration and its drivers in the Congo Basin

Michael W. Burnett[1], Gregory R. Quetin[2], Alexandra G. Konings[2]

[1]Earth Systems Program, Stanford University, Stanford, CA, USA

[2]Department of Earth System Science, Stanford University, Stanford, CA, USA

*Correspondence to*: Michael W. Burnett (mburnett@alumni.stanford.edu)

## Abstract

Evapotranspiration (ET) from tropical forests serves as a critical moisture source for regional and global climate cycles. However, the magnitude, seasonality, and interannual variability of ET in the Congo Basin remain poorly constrained due to

a scarcity of direct observations, despite it being the second-largest river basin the world and containing a vast region of tropical forest. In this study, we applied a water balance model to an array of remotely-sensed and in-situ datasets to produce monthly, basin-wide ET estimates spanning April 2002 to November 2016. Data sources include water storage changes estimated from the Gravity Recover and Climate Experiment (GRACE) satellites, in-situ measurements of river discharge, and precipitation from several remotely sensed sources. An optimal precipitation dataset was determined as a weighted average of interpolated

data by Nicholson et al. (2018), Climate Hazards Infrared Precipitation with Station Version 2 (CHIRPS2) data, and the Precipitation Estimation from Remotely Sensed Information using Artificial Neural Networks−Climate Data Record product (PERSIANN−CDR), with the relative weights based on the error magnitudes in each dataset as determined by triple collocation. The resulting water balance-derived ET ($ET_{wb}$) features a long-term average that is consistent with previous studies (117.2±3.5 cm/year), but displays greater seasonal and interannual variability than six global ET products. The seasonal cycle

of $ET_{wb}$ generally tracks that of precipitation over the basin, with the exception that $ET_{wb}$ is greater in March-April-May (MAM) than in the relatively wetter September-October-November (SON) periods. This pattern appears to be driven by seasonal variations in diffuse photosynthetically-active radiation (PAR) fraction, net radiation ($R_n$), and soil water availability. From 2002–2016, $R_n$, PAR, and vapor-pressure deficit (VPD) all increase significantly within the Congo Basin; however, no corresponding trend occurred in $ET_{wb}$. We hypothesize that the stability of $ET_{wb}$ over the study period despite sunnier and less

humid conditions is likely due to increasing atmospheric $CO_2$ concentrations that offset the impacts of rising VPD and irradiance on stomatal water use efficiency (WUE).

## 1 Introduction

The Congo River Basin in central Africa is the second-largest river basin in the world and supports one of Earth's three major humid tropical forest regions (Alsdorf et al., 2016). Approximately 24 to 39 percent of evapotranspiration (ET) from the Congo





Basin is recycled as local rainfall (Dyer et al., 2017), and model simulations indicate changes in ET within the basin affect moisture cycling across the African continent (Van Der Ent and Savenije, 2011; Bell et al., 2015; Sorí et al., 2017). Understanding the magnitude, variability, and drivers of ET in the Congo Basin is therefore crucial for studying the climate systems of central Africa and the global tropics, especially because significant environmental shifts have already been reported within the basin. For instance, deforestation is an ongoing problem in Congolese forests, with potential impacts on climate

(Laporte et al., 2007; Batra et al., 2008; Bell et al., 2015; Turubanova et al., 2018); temperatures are rising due to anthropogenic climate change (James and Washington, 2013); and many have reported a long-term decline in precipitation over the basin (Asefi-Najafabady and Saatchi, 2013; Diem et al., 2014; Zhou et al., 2014; Hua et al., 2016; Dezfuli, 2017). Such shifts are particularly concerning because Africa's tropical rainforests are already significantly drier than other humid tropical forests and exist at the threshold of conversion from evergreen to deciduous trees (Guan et al., 2015; Philippon et al., 2019).

However, the hydrology of the Congo Basin is vastly understudied relative to the region's size and influence (Alsdorf et al., 2016). In particular, no long-term observational studies of ET in the basin exist. There are also no eddy covariance towers operating within the Congo Basin. Prior studies provide only limited information and either analyze short-term (<1 month) ET observations at individual site scale (Nizinski et al., 2011, 2014) or rely on combining meteorological measurements at a site scale with local models (Bultot, 1971; Lauer, 1989; Shahin, 1994). Some studies have also used process-based models

to evaluate regional-scale ET (Matsuyama et al., 1994; Shem, 2006; Batra et al., 2008; Chishugi and Alemaw, 2009; Marshall et al., 2012; Ndehedehe et al., 2018; Crowhurst et al., 2020). However, the large size and heterogeneity of the Congo Basin render point-based approaches inadequate for basin-scale analysis of hydrological cycling, and regional scale models suffer from being poorly constrained because of the widespread lack of in-situ observations throughout the basin (e.g. little understanding of local variability in rooting depth, vegetation responses to water and light availability, canopy interception,

etc.). As a result, even basic seasonality patterns across the Congo Basin remain unclear. For example, Konings et al. (2017) showed that canopy water content increases during dry season, which could be due to either dry season leaf-out or a change in plant water uptake during the dry season that would increase ET. The latter could not be ruled out in Konings et al. (2017) due to the lack of direct ET estimates in the region.

Remote sensing offers a partial solution to this scarcity of ET observations. Remote sensing-based estimates of ET

are generally indirect, relying on physical models to link temperature, meteorological inputs, and/or other observables to the rate of ET (Zhang et al., 2016). However, these global modeling approaches are poorly constrained in the Congo River Basin and may be highly erroneous there. Alternatively, basin-scale ET can be estimated indirectly by inverting the water balance. This approach requires only three geophysical input variables: precipitation, river discharge, and the change in terrestrial water storage. Precipitation and total water storage change can both be estimated using remote sensing, with the latter determined by

gravity measurements from the Gravity Recovery and Climate Mission (GRACE) (Tapley et al., 2004; Swenson, 2012). Recent examples of this method's application include the Amazon Basin (Maeda et al., 2017; Swann and Koven, 2017) and the coterminous United States (Wan et al., 2015), as well as global examinations of basin-scale ET (Liu et al., 2016). While some previous studies have applied a similar technique to the Congo Basin as part of larger-scale studies, these studies assumed





terrestrial water storage was constant over their study periods (Marshall et al., 2012; Ukkola and Prentice, 2013; Weerasinghe

et al., 2020)—an assumption with little support. Indeed, remotely-sensed evidence suggests water storage anomalies within the basin do change significantly on monthly and interannual timescales (Crowley et al., 2006; Rodell et al., 2018). In this paper, we applied the water balance method to the Congo Basin to produce the first data-driven estimates of monthly basin-averaged ET for the period from April 2002 to November 2016. To determine the most accurate precipitation time series to use in this computation, precipitation estimated from multiple remote sensing-based approaches are combined based on

uncertainty estimation using triple collocation. We further used the resulting ET time series to explore the climatic and ecological drivers of ET seasonality and trends by comparing it against a variety of vegetation indices and meteorological drivers.

## 2 Methods

Based on mass balance, any precipitation that falls on a basin and is not removed from the basin through river discharge or ET

must increase the amount of water stored in the basin in the form of groundwater, soil moisture, or open water bodies. The equation for this mass balance can be rearranged to solve for ET as follows:

$$ET_{wb} = P - Q - \frac{dS}{dt} \tag{1}$$

where $ET_{wb}$ is monthly basin-wide evapotranspiration, $P$ is the monthly basin-wide precipitation, $Q$ is total monthly runoff from the Congo River, $S$ is the water storage anomaly within the basin expressed as an equivalent water height (Rodell et al.,

2004a, 2011), and $t$ is time. We calculate $ET_{wb}$ using $P$ from a combination of remotely-sensed and gauge-based precipitation products, as further discussed in Sects. 2.1 and 2.2. $Q$ was obtained from a stream gauge at the outlet of the Congo River at Kinshasa-Brazzaville. Lastly, $dS/dt$ was derived from the monthly change in terrestrial water storage throughout the basin, as estimated by gravitational anomaly data from GRACE (Tapley et al., 2004; Swenson, 2012).

### 2.1 Water balance data sources

The area and extent of the Congo Basin were determined using the 15-arcsecond HydroSHEDS Level 5 Basin Boundaries product (Lehner et al., 2008). The HydroSHEDS boundary produces a total basin area of 3,705,220 km$^2$—in good agreement with a recent independent estimate of 3,687,000 km$^2$ (Alsdorf et al., 2016). The HydroSHEDS product was used to trim all remotely-sensed raster data to the Congo Basin's boundaries at 0.01° spatial resolution (all datasets with coarser spatial resolutions mentioned hereafter were first resampled to 0.01° grids with no interpolation before determining their basin-wide

values).

Basin-wide runoff ($Q$) for the Congo Basin was estimated using monthly discharge data collected from the Congo River at Kinshasa-Brazzaville. The long-running gauging station is operated by the Observation Service for Geodynamical, Hydrological and Biogeochemical Control of Erosion/Alteration and Material Transport in the Amazon, Orinoco and Congo



Basins (SO-HYBAM) and captures the drainage of over 98% of the Congo Basin's area (Alsdorf et al., 2016). Because no
uncertainty estimate is available for the streamflow gauge, we assumed an uncertainty range of ±20%.

Changes in terrestrial water storage ($dS/dt$) were calculated using $S$ data from NASA's Gravity Recovery and Climate
Experiment (GRACE) satellites (Swenson and Wahr, 2006; Landerer and Swenson, 2012; Swenson, 2012). In order to estimate
monthly $S$, three independent GRACE solutions in 1° grids from Geoforschungs Zentrum Potsdam (GFZ), Jet Propulsion
Laboratory (JPL), and the Center for Space Research at University of Texas, Austin (CSR) were retrieved. A scale factor grid
was also applied to the GRACE data to account for attenuation of small-scale surface mass variations (Landerer and Swenson,
2012). The arithmetic mean of the three $S$ solutions was used in the primary $dS/dt$ calculation in order to reduce noise (Wahr
et al., 2006; Sakumura et al., 2014), though all three independent $S$ solutions were also used to calculate unique $dS/dt$ values
in order to estimate uncertainty in the GRACE products (Lee et al., 2011). The $S$ data were converted to $dS/dt$ values using a
centered-difference approach at monthly timescale:

$$\frac{dS}{dt}_n = (S_{n+1} - S_{n-1}) \tag{2}$$

where the $S$ terms are expressed in centimeters of equivalent water height averaged over the entire Congo Basin for the months
before and after month $n$ (Landerer et al., 2010). The uncertainty of $dS/dt$ is calculated as half of the difference between the
highest and lowest $dS/dt$ values from the three GRACE $S$ solutions in any given month (Lee et al., 2011)

Beginning in early 2011, the GRACE mission began an active battery management strategy that resulted in data gaps
every several months. In order to reconstruct $dS/dt$ data from 2011–2016, we use its average seasonal cycle to correct for
interannual variability based on the amount of that variability in nearby months. First, the mean monthly cycle of $S$ was
calculated from data-complete months from 2002–2016. For every missing month from 2011–2016, the average $S$ from the
two other months in the same season of the same year (DJF, MAM, JJA, and SON) was compared to the corresponding value
from the multi-year $S$ means. The resulting ratio was then multiplied by the multi-year $S$ mean of the missing month to create
the reconstructed $S$ value. Because the sum of multi-year mean $S$ values from October and November is nearly equal to zero
and consequently produces unrealistically-scaled values for September, missing September values were instead interpolated
using August and October of the same year. Repeating the same procedure for months that are available in the GRACE dataset
(i.e. calculating what the reconstructed value would be if it were not available, and comparing it to the observations) shows
that this seasonal-scaling interpolation reproduces true $S$ fairly accurately: from 2002–2016, each of the twelve months was
reconstructed with a mean R² of 0.75 and a mean root mean square error (RMSE) of 2.80 cm (relative to average seasonal $S$
variations of ~10 cm). Applying this procedure to the mean $S$ data from the three monthly GRACE solutions produced the
complete $dS/dt$ time series that determined the study period for our water balance model (4/2002–11/2016).

Due to the uncertainty of precipitation ($P$) estimates in the Congo (Washington et al., 2013), $P$ was estimated using
an array of five datasets with different methodologies. These five datasets were chosen because recent validation efforts have
shown them to be the most accurate for the Congo Basin (Nicholson et al., 2018, 2019). They include gridded precipitation
data from the Global Precipitation Climatology Centre (GPCC) Version 7.0, which uses interpolation with a worldwide





network of rain gauges to produce monthly precipitation grids (Schneider et al., 2015). GPCC-compiled gauges within the Congo are extremely sparse after 2000 (Nicholson et al., 2019), and the GPCC version 7.0 product only lasts through 2013. The Tropical Rainfall Measuring Mission (TRMM) 3B43 Version 7 product (also known as TRMM Multi-Satellite
Precipitation Analysis or TMPA), which consists of monthly mean precipitation rate grids, is generated using microwave and infrared sensors on TRMM and other satellites as well as gauge data from the GPCC (Huffman et al., 2007). The Precipitation Estimation from Remotely Sensed Information using Artificial Neural Networks: Climate Data Record (PERSIANN-CDR) product uses satellite infrared data and a neural network approach, calibrated with precipitation forecasts, microwave data, and GPCC gauge data, to produce grids of precipitation estimates at the daily timescale (Ashouri et al., 2015). While PERSIANN
is also available without the GPCC gauge corrections that make PERSIANN-CDR so similar to GPCC v7 and TRMM 3B43 over the Congo Basin (Nguyen et al., 2018), it was not used here because it severely overestimates $P$ across Africa (Beighley et al., 2011; Thiemig et al., 2012). Recent studies have found that TRMM Version 7 3B43 and PERSIANN-CDR both perform reasonably well over central Africa (Munzimi et al., 2015; Awange et al., 2016).

Notably, the above three products all depend on GPCC rain gauges to some degree. As a result, all three datasets
feature similar rainfall trends and model performance over the Congo Basin during the 2002–2016 period studied here and cannot be considered truly independent $P$ datasets (Nicholson et al., 2019). To that end, the Climate Hazards group Infrared Precipitation with Stations Version 2.0 (CHIRPS2) product, which uses two thermal infrared datasets and interpolated gauge data, was also included as a more independent dataset (Funk et al., 2015). While CHIRPS2 does incorporate some gauge data that overlap with the GPCC product, it is not scaled to fit GPCC data to the same extent as TRMM 3B43 or PERSIANN-CDR
are over central Africa (Nicholson et al., 2019)—in fact, the near-total lack of rain gauges within the basin in both CHIRPS2 and GPCC leads to a low correlation between the two datasets within the study area (Funk et al., 2015), indicating the $P$ datasets maintain a high degree of independence. Finally, a recent gauge-based dataset developed for the Congo Basin, NIC131-gridded, served as another independent precipitation data source with coverage through 2014 (Nicholson et al., 2018). The monthly 2.5° NIC131-gridded product was created by applying a spatial reconstruction technique based on principal
component analysis to a gauge network that, due to the severe decline of GPCC coverage during the 1990s, is largely independent of the GPCC's gauges in Africa (Nicholson et al., 2019).

## 2.2 Comparing and unifying precipitation estimates

Because the above five datasets each individually remain highly uncertain, and because no accurate independent basin-wide validation is possible, triple collocation (TC) was used to estimate the error statistics of the different datasets, and ultimately
combine them by weighting them according to their relative errors. TC is a method for characterizing systematic and random errors in geophysical measurements using three independent, collocated time series, and even if these datasets are individually noisy (Stoffelen, 1998; McColl et al., 2014). It is particularly valuable in gauge-sparse regions like the Congo Basin because it does not rely on independent error-free validation data. TC-based error calculations have previously been used in a wide variety of geophysical settings—among others, TC was recently used to determine the relative weightings of different





hydrologic flux estimates in a neural network-based data combination effort in a manner conceptually analogous to its use here (Alemohammad et al., 2017). Rather than the linear model used in most TC applications, we used a multiplicative model that is more appropriate for quantifying errors in precipitation estimates (Alemohammad et al., 2015). In the multiplicative error model, true precipitation rate $T$ is assumed to be related to the estimated precipitation of product $i$, $P_i$, as follows:

$$P_i = a_i T^{\beta_i} e^{\epsilon_i} \tag{3}$$

in which $a_i$ is the multiplicative error, $\beta_i$ is the deformation error, and $\epsilon_i$ is the random residual error (which is assumed to have zero-mean).

Assuming the three collocated precipitation estimates' residual errors are uncorrelated with each other, and are uncorrelated with the true precipitation values, the RMSEs of all three input $P$ datasets may be calculated with Eqs. (4–6):

$$\sigma_{p_1}^2 = C_{11} - \frac{C_{12}C_{13}}{C_{23}} \tag{4}$$

$$\sigma_{p_2}^2 = C_{22} - \frac{C_{12}C_{23}}{C_{13}} \tag{5}$$

$$\sigma_{p_3}^2 = C_{33} - \frac{C_{13}C_{23}}{C_{12}} \tag{6}$$

where $C_{ij}$ is the $(i, j)$th element of the sample covariance matrix between the three log-transformed datasets and $\sigma_{pi}$ is the RMSE of the log-transformed $P_i$ time series. $\sigma_{pi}$ can be converted to the actual RMSE of $P_i$ by multiplying by the mean value of $P_i$ (Alemohammad et al., 2015).

The errors of the five $P$ datasets were evaluated by applying TC to triplets of products deemed relatively independent. That is, TC was repeated three times using different triplets: TRMM–NIC131–CHIRPS2, GPCC–NIC131–CHIRPS2, and PERSIANN–NIC131–CHIRPS2. The three RMSEs calculated for NIC131-gridded and CHIRPS2 were then averaged and compared to the RMSEs calculated for TRMM, GPCC, and PERSIANN-CDR. In order to combine the most accurate $P$ time series (and their estimated errors) into a single unified $P$ estimate, weighting factors were assigned to each time series in a 180 manner inversely proportional to the product RMSE. That is, each weighting factor $w_i$ was assigned as in (Eq. 7):

$$w_i = \frac{RMSE_i^{-1}}{\sum_1^3 RMSE_i^{-1}} \tag{7}$$

The best-estimate rate of precipitation for each month was then calculated as a weighted average across all five original precipitation products, using $w_i$. The resulting dataset's RMSE was also used to propagate precipitation uncertainty into the uncertainty of $ET_{wb}$ using a root-mean-square sum of the weighted errors. While longer time series are generally preferred for 185 use in TC in order to reduce sampling error, data prior to 2002 were discarded in the TC analysis because the greater number of gauges prior to 2002 likely leads to different error statistics than in this period (Nicholson et al., 2018).

## 2.3 Comparison to global ET products

Many hydrological studies of the Congo Basin rely on global ET products to constrain their models (e.g. Ndehedehe et al. 2018; Hassan and Jin 2016). We analyzed six widely-used global ET data products and evaluated their performance relative 190 to $ET_{wb}$. MOD16A2 Version 5 is a global ET data product based on the Penman-Monteith equation, meteorological reanalysis,




and remotely-sensed land surface data from the Moderate Resolution Imaging Spectroradiometer (MODIS) satellite mission (Mu et al., 2013). The Global Land Evaporation: Amsterdam Model Version 3.1a (GLEAM v3.1a) product estimates Priestley-Taylor potential ET (PET) from reanalysis radiation and temperature data, then reduces PET to actual ET using remotely sensed soil moisture and vegetation optical depth measurements (Miralles et al., 2011; Martens et al., 2017). Modern-Era

Retrospective Analysis for Research and Applications, Version 2 (MERRA-2) is a reanalysis product that integrates a wide variety of observation types from satellites and in-situ sources to produce terrestrial ET estimates using a water balance approach (Gelaro et al., 2017). The Global Land Data Assimilation System Version 2.1 Noah (GLDAS-Noah) product is a land surface simulation forced by a combination of model and observation datasets that provides monthly mean ET estimates (Rodell et al., 2004b).

Lastly, two global ET products based on upscaling tower data from global FLUXNET eddy covariance network (Baldocchi et al., 2002) were included: the Model Tree Ensemble (FLUXNET-MTE) product uses a tree-based machine learning approach to upscale carbon, water, and energy flux observations using external global data sources, resulting in a monthly 0.5° global dataset (Jung et al., 2011). The more recent FLUXCOM product uses machine learning algorithms and additional time-varying meteorological inputs to achieve greater accuracy in upscaling flux tower data (Jung et al., 2019). This

study uses FLUXCOM's daily RS+METEO version because of its lower ET uncertainty in Africa (Jung et al., 2019). However, it should be noted that FLUXNET-MTE and FLUXCOM, like the physical modeling approaches above, have primarily been validated against observational data in the mid-latitudes. There are no FLUXNET towers located within the Congo basin that could have been used for training these and other models. All datasets were averaged across the Congo basin using linear interpolation.

The accuracies of these six products were evaluated by comparing them to the monthly $ET_{wb}$ values: RMSEs, Pearson correlation coefficients, and Taylor skill scores were calculated for each dataset versus $ET_{wb}$. Only the years 2002–2011 are common to all six ET datasets and the GRACE $S$ data, so all statistics were calculated over this period. Pearson correlation coefficients help determine the ability of each ET model to predict $ET_{wb}$, while Taylor skill scores allow a comparison of the variability present in each model by accounting for their standard deviations (Taylor, 2001). The average seasonal cycles and

interannual variations of the products are also compared to better understand similarities and differences between the products.

**2.4 Meteorological and vegetation data**

To examine potential drivers of ET's seasonality, interannual variability, and long-term trends in the Congo Basin, $ET_{wb}$ is compared to a host of meteorological and vegetation data including photosynthetically-active radiation (PAR), net radiation ($R_n$), vapor-pressure deficit (VPD), air and skin temperatures ($T_a$ and $T_s$), solar-induced fluorescence (SIF), and enhanced

vegetation index (EVI). We used all-sky monthly mean PAR and $R_n$ data from the Clouds and the Earth's Radiant Energy System (CERES) project's 1° gridded products. PAR data were derived from the synoptic surface flux model (SYN1deg) (Doelling, 2017), which divides surface PAR fluxes into direct (PAR$_{dir}$) and diffuse (PAR$_{diff}$) components, while $R_n$ data were derived from the Energy Balanced and Filled (EBAF) climate data record (Loeb, 2017). The global reanalysis model ERA-





Interim (Dee et al., 2011) provided surface air temperature and relative humidity data in 6-hour increments, which were used
to calculate monthly VPD means of the entire basin using linear interpolation. Although reanalysis models over Central Africa
remain uncertain and poorly constrained (Lorenz and Kunstmann, 2012; Brands et al., 2013), these VPD values were tested
against hourly VPD data from Automated Surface Observing Systems (ASOS) and Met Office Integrated Data Archive System
(MIDAS) weather reports from the Congo Basin (Met Office, 2012) and were found to capture monthly cycles of VPD with
acceptable accuracy (Fig. S1).

Monthly mean $T_a$ and $T_s$ from the Congo Basin were sourced from the ascending (daytime) retrievals of the
Atmospheric Infrared Sounder (AIRS) Level 3 monthly product (Kahn et al., 2014). The 740 nm SIF data from the Global
Ozone Monitoring Experiment 2 (GOME-2) platform were retrieved from the GOME2_Fluorescence Version 26 Level 3
dataset (Joiner et al., 2013). The GOME-2 SIF dataset is known to have suffered from a significant sensor decay problem
resulting in a spurious worldwide downward trend, so the SIF data were not used in any long-term trend analyses (Zhang et
al., 2018). MODIS Collection 6 EVI data processed with the Multi-Angle Implementation of Atmospheric Correction
(MAIAC) algorithm were converted to monthly means from 8-day composite rasters (Huete et al., 2002; Lyapustin et al.,
2018). The MAIAC algorithm, which eliminates errors from aerosols and sun-sensor geometry issues in MODIS data, has
previously proven beneficial for examining vegetation greenness in tropical forests (Hilker et al., 2012; Lyapustin et al., 2012;
Bi et al., 2016). Lastly, annual land cover data from the MODIS-based MCD12C1 Version 6 dataset were retrieved for 2002–
2016 and modally-averaged to produce a single land cover classification of the Congo Basin (Friedl and Sulla-Menashe, 2015).
Pixels were aggregated into dominantly deciduous or evergreen vegetation types according to the International Geosphere-
Biosphere Programme's (IGBP) 17-class land cover scheme, with savannas and grasslands considered deciduous and
permanent wetlands considered evergreen. Other vegetation types that are more difficult to generalize (e.g. croplands, mixed
forests, and shrublands) were spatially-limited enough to be ignored once the land cover data were majority-resampled to
match the 1° pixel size of our study's coarsest datasets (Fig. S2).

## 2.5 Removing seasonal cycles and long-term trends

In order to track correlations between $ET_{wb}$, meteorological variables, and vegetation indices, the Breaks for Additive Season
and Trend (BFAST) R package (Verbesselt et al., 2015) was used to search for abrupt changes in the trends of our time series,
identify linear long-term trends, and remove the average seasonal cycle from the data (Verbesselt et al., 2010a, 2010b).

## 250 3 Results

### 3.1 Triple collocation of precipitation datasets

The results of the TC analysis are provided in Table 1. NIC131-gridded exhibited the lowest RMSE in all three
triplets, from 0.60 cm/month to 0.72 cm/month (mean 0.65 cm/month) depending on the triplet. The TC results indicate that
NIC131-gridded, a Congo-specific gauge-based dataset designed using meteorological stations absent from the GPCC network



and a principal component-based statistical approach, is the best currently-available *P* dataset for the Congo Basin after 2002 (Nicholson et al., 2018). The results in Table 1 agree well with those of Nicholson et al. (2019), which found CHIRPS2 and PERSIANN-CDR to be more accurate than TRMM and GPCC in the Congo Basin after 1998. Prior work has also demonstrated that CHIRPS2 is among the best *P* products available for central Africa and outperforms TRMM and PERSIANN-CDR on a monthly basis (Dembélé and Zwart, 2016; Nicholson et al., 2019), consistent with the results in Table

1. Given the decreasing availability of Congolese rain gauge data in the GPCC database and the difficulty of measuring *P* with satellites in central Africa (McCollum et al., 2000; Yin and Gruber, 2010; Awange et al., 2016; Nicholson et al., 2018), it is not surprising that the GPCC-based products generally displayed higher errors.

**Table 1: Root mean square errors (RMSEs) for the five P datasets (in three triplets) evaluated in this study, as well as the weighting factors used to unify the three most accurate datasets. All values are in in units of cm/month.**

| Dataset | RMSE Triplet 1 | RMSE Triplet 2 | RMSE Triplet 3 | Mean RMSE | Weighting Factor |
|---|---|---|---|---|---|
| TRMM 3B43 | 1.67 | - | - | 1.67 | - |
| GPCC Version 7 | - | 1.66 | - | 1.66 | - |
| PERSIANN-CDR | - | - | 1.60 | 1.60 | 0.19 |
| NIC131-gridded | 0.65 | 0.72 | 0.60 | 0.65 | 0.47 |
| CHIRPS2 | 0.93 | 0.88 | 0.96 | 0.93 | 0.33 |


TRMM, GPCC, and PERSIANN-CDR—which all integrate GPCC rain gauges in some capacity—are highly correlated and therefore feature similar RMSEs between 1.60 cm/month (PERSIANN-CDR) and 1.67 cm/month (TRMM). Therefore, our subsequent analyses discard GPCC and TRMM and use only PERSIANN-CDR—the most accurate of the three GPCC gauge-related datasets. As discussed in Sect. 2.2, PERSIANN-CDR was implemented in a weighted average in

combination with NIC131-gridded and CHRIPS2 to create a unified *P* time series, $P_{TC}$. The NIC131-gridded only lasts through 2014, so from 2015–2016 only CHIRPS2 and PERSIANN-CDR were used in $P_{TC}$. The uncertainty in $P_{TC}$ was estimated to be 0.30 cm/month from 2002–2014 and 0.59 cm/month from 2015-2016 (after NIC131-gridded data coverage ends)—both lower than the RMSEs of any of the individual *P* products tested.

### 3.2 Water balance ET estimates

In Fig. 1, clear seasonal cycles as well as interannual variations are visible in all four of the hydrologic fluxes from Eq. 1: the rainy MAM and SON seasons show local peaks in $ET_{wb}$ as well as $dS/dt$, which is generally a negative flux (representing water leaving the land surface system) for most of the rest of the year. *Q* has the least temporal variability of the fluxes and is the smallest in magnitude, although it exhibits increased runoff 1–2 months after the primary SON rainy season. Mean annual $ET_{wb}$ is 117.2±3.5 cm/year (calculated from 2003–2015, the data-complete years of the study period). Mean annual $P_{TC}$ from

2003–2015 is 150.4±2.6 cm/year, and mean annual *Q* is 33.7 cm/year. The *dS/dt*, which fluctuates between positive and negative values, ranges from −3.2 to 3.7 cm/month on average.



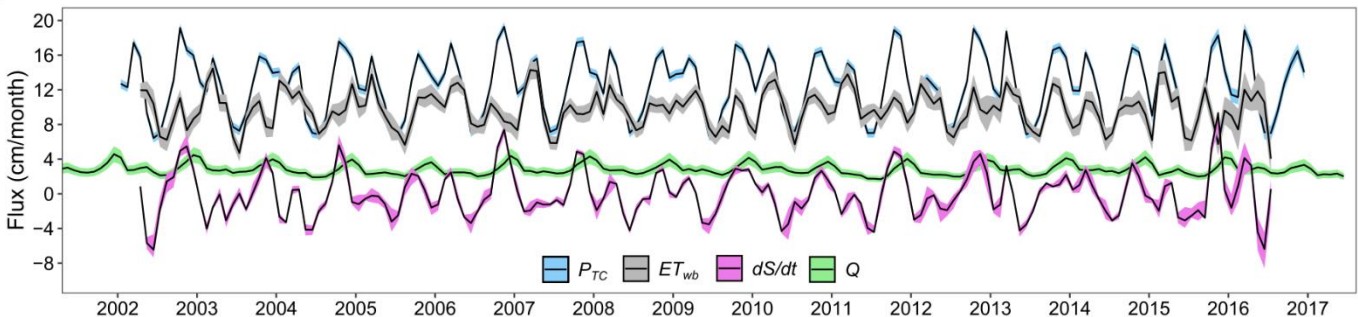

**Figure 1: Time series of the four water balance components from 2002-2016. Data are monthly, basin-wide averages in cm water height equivalents. Black lines represent mean values; ribbons represent uncertainty ranges.**

Plotting monthly means of the water balance fluxes provides further clarity regarding their seasonal cycles (Fig. 2). October and November are the rainiest months, followed by March and April, while June and July are the driest months of the year. The $S$ regenerates mostly during the very wet October and November months and less so during December, the secondary rainy season in March and April, and in September with the onset of the primary rainy season. $S$ loses water fastest during May and June, reaching its minimum during June, when $ET_{wb}$ exceeds $P_{TC}$ on average (Matsuyama et al., 1994). Interestingly, while $ET_{wb}$ tracks the seasonality of $P_{TC}$ to an extent, it peaks in March during the secondary rainy season rather than during the primary, wetter, SON wet season. The possible causes of this difference between precipitation and ET seasonality are analyzed further in Sect. 4.2.

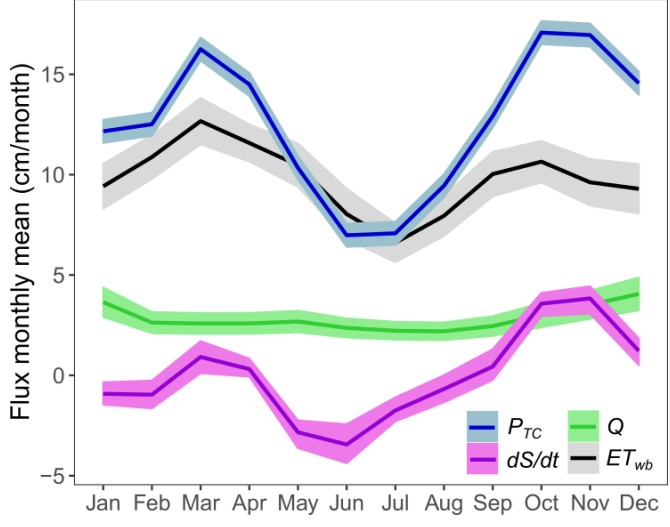

**Figure 2: Mean monthly cycle of the four water balance components from 2002-2016. Dark lines represent mean values; ribbons represent uncertainties.**





Table 2 summarizes fourteen mean annual ET estimates from the Congo Basin found in the literature. The studies produce a mean ET of 116.7 cm/year with a standard deviation of 6.8 cm/year and a median ET of 118.9 cm/year, although different study periods and a variety of methods were used to estimate actual ET. All but one historical ET estimate fall within 10% of mean annual $ET_{wb}$, showing good agreement between the present study's ET estimates and prior literature on the
subject.

**Table 2: Historical estimates of mean annual basin-wide ET from the literature. Mean and median values are derived from the literature and presented alongside the mean annual $ET_{wb}$ from this study.**

| Source | Mean ET (cm/yr) | Time span |
|---|---|---|
| Balek 1977 | 124.8 | climatology |
| Balek 1983 | 122.4 | climatology |
| Bricquet 1988 | 123.0 | climatology |
| Bultot 1971 | 119.6 | climatology |
| Chishugi and Alemaw 2009 | 109.8 | 1961-1990 |
| Matsuyama et al. 1994 | 125.0 | 1985-1988 |
| Nicholson et al. 1997 | 112.7 | climatology |
| Oki et al. 1993 | 120 | 1985-1988 |
| Olivry et al. 1993 | 108.6 | 1951-1990 |
| Pan et al. 2012 | ~102 | 1984-2006 |
| Pinet and Souriau 1988 | 118.2 | climatology |
| Russell and Miller 1990 | 114 | climatology |
| Shem 2006 | 122.3 | 1979-1994 |
| Ukkola and Prentice 2013 | ~111 | 1963-1998 |
| Mean: | 116.7 | |
| Median: | 118.9 | |
| This study | 117.2±3.5 | 2003-2015 |

**3.3 Comparing the $ET_{wb}$ seasonal cycle to global ET models**

The seasonal cycle of $ET_{wb}$ is compared to those of six global ET products in Fig. 3. Multiple datasets generally disagree with the magnitude of basin-wide $ET_{wb}$. Indeed, MOD16A2 and GLEAM v3.1a fall outside the uncertainty range of $ET_{wb}$ more often than not. GLDAS-Noah and FLUXCOM monthly means both fall within the uncertainty range of $ET_{wb}$ eight months out of the year, while MERRA2 does so seven months out of the year. While all models display the basin's low JJA ET to some degree, they generally fail to capture the fast recovery of $ET_{wb}$ from August–September. Most models also correctly find ET
to peak during the MAM rainy season (Matsuyama et al., 1994; Pan et al., 2012; Crowhurst et al., 2020), but all global ET products underestimate how much larger the MAM $ET_{wb}$ peak is than the SON one. For instance, FLUXNET-MTE plots SON ET as roughly equivalent to MAM ET. The peak ET months for each rainy season also lag those of $ET_{wb}$ for several products.





In general, the global ET products fail to capture the magnitude of seasonal variations in $ET_{wb}$, although some track $ET_{wb}$ much more closely than others.

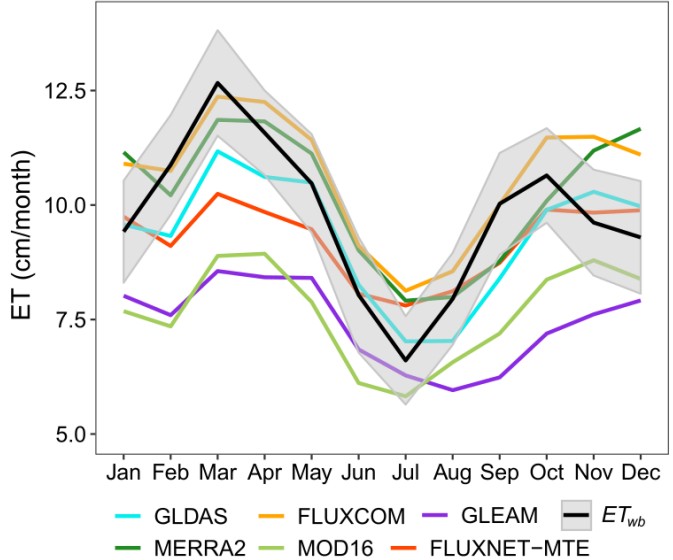

**Figure 3: Mean monthly cycle of $ET_{wb}$ plotted alongside the mean monthly cycles of six global ET products. The grey ribbon represents $ET_{wb}$ uncertainty.**

Global ET products are evaluated against $ET_{wb}$ from 2003–2011 using several metrics in Table 3. Mean annual ET ranges from 88.7 cm/year (GLEAM v3.1a) to 127.6 cm/year (FLUXCOM), compared to the $ET_{wb}$ annual mean of 118.0±3.5 cm/year for that time period. Excepting GLEAM v3.1a and MOD16A2, the global ET product annual averages all fall within 10% of $ET_{wb}$'s. FLUXCOM produces the highest Pearson correlation coefficient versus $ET_{wb}$ while MERRA2 produces the lowest, and FLUXCOM again leads all products in Taylor skill score while FLUXNET-MTE demonstrates the lowest. FLUXCOM also features the lowest RMSE relative to $ET_{wb}$; GLEAM v3.1a features the highest. GLEAM v3.1a underestimates mean annual $ET_{wb}$ by nearly 25%, exhibits a low Taylor score, and has the highest RMSE of all six products. MOD16A2 has the second-highest RMSE of the six products evaluated and generally underestimates the magnitude and seasonality of $ET_{wb}$ in the Congo Basin while achieving a correlation coefficient of only 0.54 and a Taylor score of 0.58.





**Table 3: Mean annual ETs from 2003-2011 alongside Pearson correlation coefficients, Taylor skill scores, RMSEs, and standard deviations from 2002-2011 for six global ET products in comparison to $ET_{wb}$.**

| ET product | Mean annual ET (cm) | Pearson correlation coefficient | Taylor skill score | RMSE (cm) | Standard Deviation |
|---|---|---|---|---|---|
| $ET_{wb}$ | 118.0±3.5 | | | | 2.1 |
| MOD16A2 | 94.2 | 0.54 | 0.58 | 2.7 | 1.3 |
| GLEAM | 88.7 | 0.53 | 0.40 | 3.0 | 0.9 |
| FLUXNET-MTE | 111.1 | 0.63 | 0.36 | 1.8 | 0.8 |
| FLUXCOM | 127.6 | 0.74 | 0.71 | 1.6 | 1.3 |
| GLDAS-Noah | 110.7 | 0.64 | 0.69 | 1.7 | 1.4 |
| MERRA2 | 119.7 | 0.43 | 0.68 | 2.1 | 1.7 |

**3.4 Drivers of ET seasonality and variability**

As discussed in Sec 3.2, the shape of $ET_{wb}$'s seasonal cycle roughly follows that of $P_{TC}$, since water availability and vegetation
productivity modulate ET. However, $ET_{wb}$ is greater during the MAM rainy season than in the SON rainy season, despite the
latter being wetter than the former. On average, $ET_{wb}$ also exceeds $P_{TC}$ during June. These findings are consistent with previous
studies that found basin-wide ET can peak during MAM (Matsuyama et al., 1994; Pan et al., 2012; Crowhurst et al., 2020),
although the drivers behind this seasonal cycle are less clear. To help develop hypotheses on the nature of ET's drivers, monthly
mean $ET_{wb}$ is compared to several climatic drivers and indices reflecting seasonally varying vegetation activity (Fig. 4).
335        Possible environmental drivers of the $ET_{wb}$ seasonality include soil water availability, water demand from VPD,
radiation, and temperature. GRACE-derived $S$ can be assumed to be a partial proxy for water availability (though note that not
all water measured by $S$ is necessarily accessible to plant roots or available for soil evaporation; see Sect. 4.2.4). $S$ is
significantly lower during SON than MAM (Fig. 4f), consistent with the relatively lower SON $ET_{wb}$. The relatively lower SON
$S$ could be due to the much lower precipitation during JJA than DJF (Fig. 3) and/or due to a difference in how much of the
rainfall infiltrates the land surface. VPD is fairly low in both wet seasons, although still elevated in September following the
JJA dry season (Fig. 4e). $R_n$ is lower during the SON wet season than the MAM, likely contributing to the lower $ET_{wb}$ in SON.



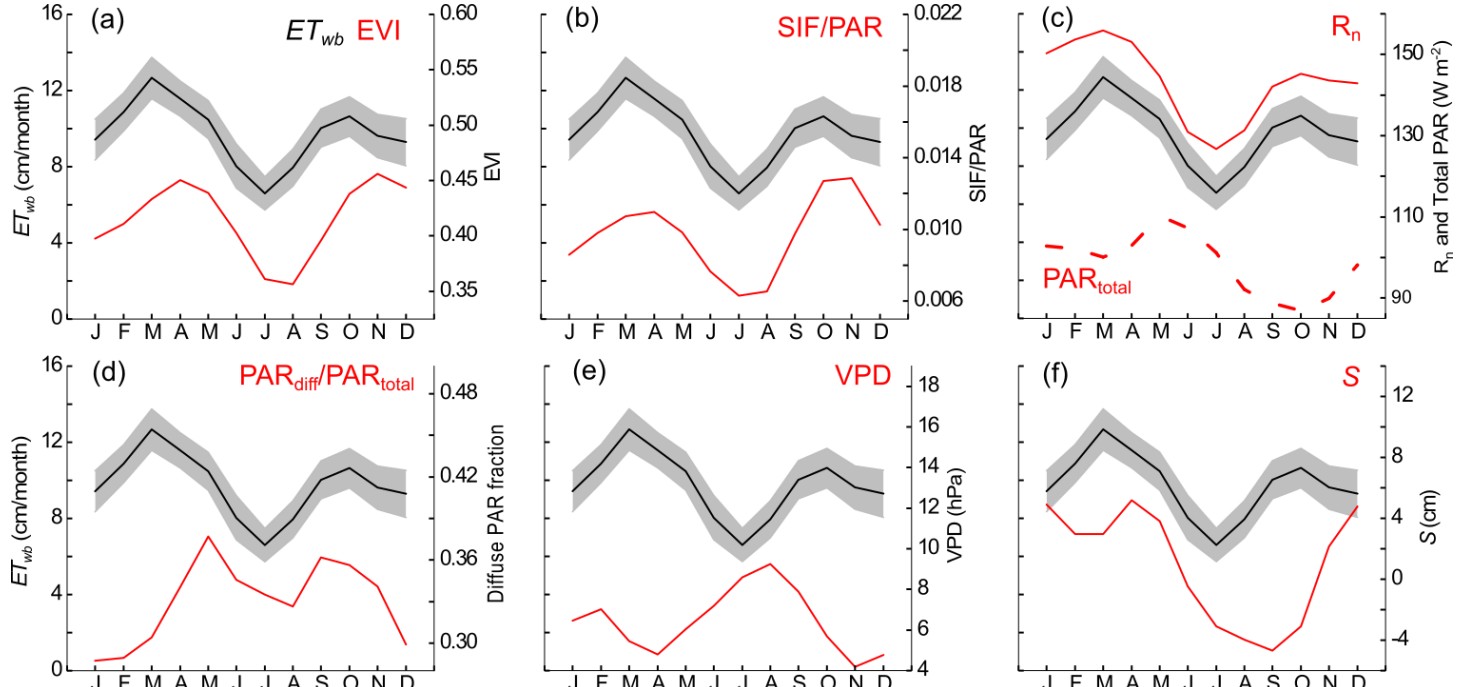

**Figure 4: Mean monthly cycle of $ET_{wb}$ (black line with gray uncertainty range) plotted alongside those of (a) MAIAC-processed EVI, (b) SIF/PAR, (c) $R_n$ and total PAR, (d) Diffuse PAR fraction ($PAR_{diff}/PAR_{total}$), (e) VPD, and (f) $S$ from GRACE (red lines). Data are averaged over the entire basin area. Note that the mean $ET_{wb}$ curve and scale is the same in each sub-panel.**

The variability of ET is expected to be linked to vegetation phenology through the large contribution of transpiration to overall ET in the densely vegetated Congo Basin (Lian et al., 2018). However, both MAIAC EVI (Fig. 4a) and PAR-normalized SIF (Fig. 4b) show greater vegetation greenness and photosynthesis, respectively, during the SON wet season than during the MAM wet season. This suggests relatively greater water use efficiency (WUE) in SON and/or a relatively greater contribution of direct soil/canopy evaporation to ET in MAM (although bare soil evaporation is expected to be a minority of total ET in the densely-forested basin; see Sect. 4.2). This greater water use efficiency during the SON than the MAM season could also be driven by a relatively greater ratio of diffuse PAR to total PAR during SON (Fig. 4d) which can increase photosynthetic efficiency (Mercado et al., 2009), as further discussed in Sect. 4.2.3.

The basin-wide analyses in Fig. 4 are almost certainly masking significant sub-basin variability. The sub-basin division of ET is not known, and dividing the coarse-resolution $S$ at the sub-basin level is also highly uncertain. Nevertheless, we considered the sub-basin variation of select productivity and climatic metrics (Fig. 5). The basin was divided into the equatorial evergreen forest (which straddles the equator), Northern deciduous ecosystems, and Southern deciduous ecosystems (Fig. S2). The deciduous regions feature larger seasonal variations in all three variables than the evergreen forest (Fig. 5), but the opposite seasonalities of the Northern and Southern regions partially offset one another and produce basin-wide EVI and SIF/PAR cycles that are broadly similar to those of the evergreen forest (Figs. 4a, b; 5b). The greater extent of the Southern





deciduous region results in basin-wide EVI and SIF/PAR minima during JJA rather than DJF, and basin-averaged VPD likewise peaks during JJA (Fig. 5e) despite its low variability in the extensive evergreen forest region (Fig. 5b). Taken together, the results of Fig. 5 suggest that a basin-wide analysis is informative despite averaging over multiple vegetation types.

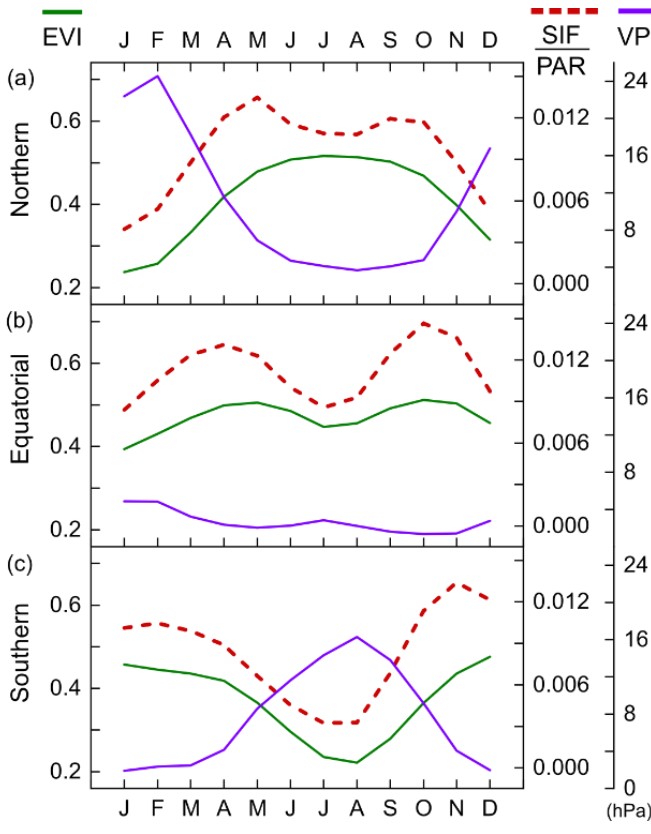

**Figure 5: (a) Average monthly cycles of MAIAC-processed EVI (green line), SIF/PAR (red dashed line), and VPD (purple line) for the northern deciduous area of the Congo Basin. (b)-(c) As with (a), but for the equatorial evergreen and southern deciduous regions, respectively. Scales are consistent between plots (a)-(c) for each variable.**

## 3.5 Long-term climatic shifts and their impacts in the Congo Basin

We detect no significant linear trends in $ET_{wb}$, $P_{TC}$, $dS/dt$, or $Q$ from 2002–2016 after removing average seasonal cycles with BFAST (Fig. 6). However, several interannual trends are detectable in other environmental data (Fig. 7). PAR, $R_n$, and VPD all increase significantly from 2002–2016 after the average seasonal cycle is removed from the time series, indicating the Congo Basin has become sunnier and less humid in recent years. This progression to sunnier and less humid conditions in the Congo Basin is not reflected in $ET_{wb}$ and productivity (as measured by MAIAC EVI), which do not show long-term changes over the past two decades (Figs. 6d, 7d).





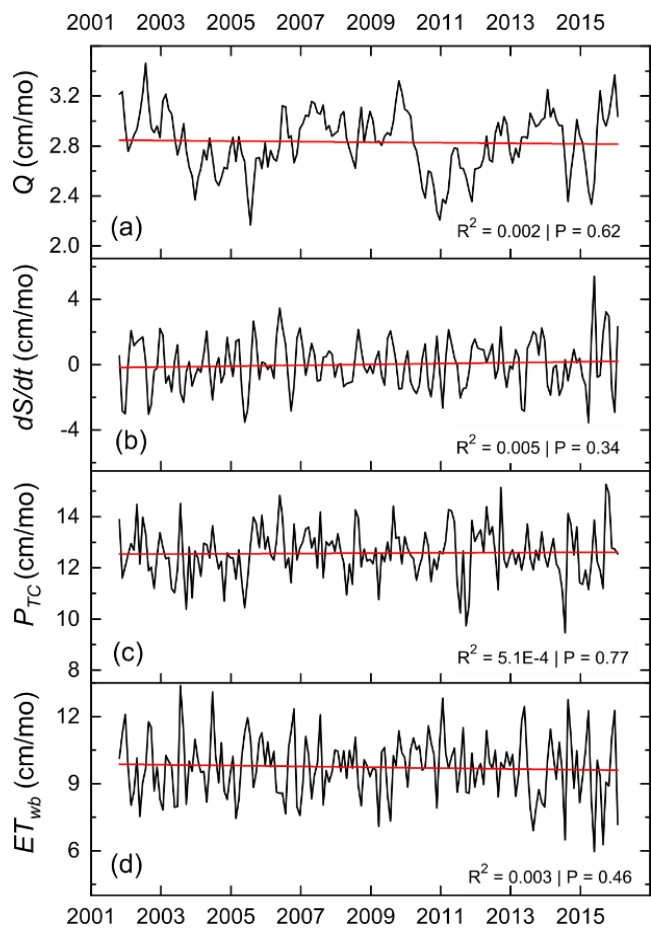

Figure 6: Linear regressions of deseasonalized monthly (a) $Q$, (b) $dS/dt$, (c) $P_{TC}$, and (d) $ET_{wb}$.



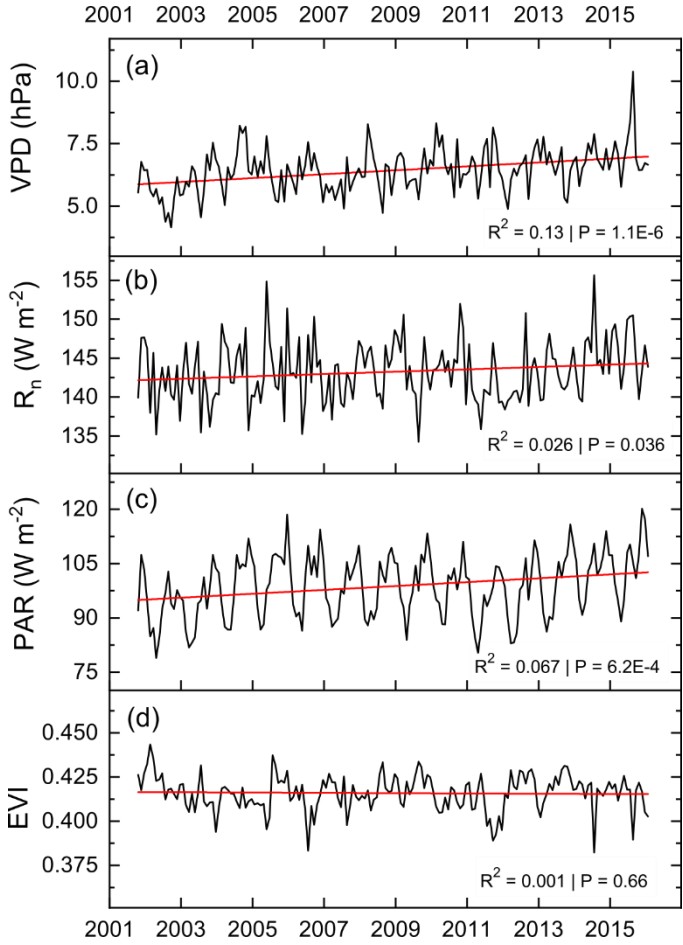

**Figure 7: Linear regressions of deseasonalized monthly (a) VPD, (b) $R_n$, (c) $PAR_{total}$, and (d) MAIAC EVI.**

## 4 Discussion

### 4.1 The value of water balance-based ET estimation in the Congo Basin

The scarcity of operational precipitation gauges and complete lack of eddy covariance towers within the Congo river basin have previously restricted ET estimates to process-based models, short-term ET observations at site scale, and global products with insufficient validation in tropical Africa. The water balance-based $ET_{wb}$ derived here provides a basin-wide constraint on ET. It has an uncertainty that is relatively low compared to its average seasonal cycle (Fig. 2), and its magnitude matches well with previous long-term ET estimates from the basin (Table 2). The shape of its annual seasonal cycle (with $ET_{wb}$ peaking in

MAM rather than in SON) also agrees with several previous ET modeling efforts in the basin (Matsuyama et al., 1994; Pan et al., 2012; Crowhurst et al., 2020). These findings support the accuracy of the water balance model in estimating basin-wide



ET, and indicate the combination of $P$ products used is relatively accurate when considered across the basin and aggregated using triple collocation.

## 4.2 The seasonal imbalance of $ET_{wb}$ and $P$ maxima

As expected, the seasonal cycle of $ET_{wb}$ mostly follows that of precipitation, with two annual dry and two annual wet seasons. However, the seasonal cycles of $P$ and $ET_{wb}$ feature an interesting offset: whereas SON is the wetter of the two rainy seasons, $ET_{wb}$ is greater during MAM than during SON (Fig. 2). The $ET_{wb}$ peak during MAM is supported by several previous modeling efforts with a variety of methodologies (Matsuyama et al., 1994; Pan et al., 2012; Crowhurst et al., 2020), indicating it is not simply an artifact of the water balance model or the data sources used here. Yet the underlying causes of the imbalance are not

well understood (Crowhurst et al., 2020). We evaluated several possible drivers of the observed MAM peak in $ET_{wb}$ including phenology, photosynthetic production (via EVI and SIF), terrestrial water availability (via GRACE-derived $S$), PAR, $R_n$, VPD, $T_a$, and $T_s$. Ultimately, we conclude that the availability of radiative energy (especially photosynthetically-favorable diffuse radiation) and greater soil water availability during MAM provides the most likely explanation for its high ET rates.

   As transpiration is the dominant component of ET throughout the Congo Basin (Lian et al., 2018), primary

productivity (through its links to stomatal closure and active leaf area) likely explains the seasonality of $ET_{wb}$ to some degree. In fact, recent studies have found that evergreen forests in equatorial Africa exhibit a similar greenness seasonality to the seasonality of $ET_{wb}$, with a bimodal cycle generally aligning with $P$ but peaking in MAM instead of the wetter SON (Betbeder et al., 2014; Philippon et al., 2016). However, these studies used MODIS products that did not fully account for sun-sensor geometry and other sources of error at low latitudes. To this end, the MAIAC algorithm (Lyapustin et al., 2012) reduces noise

and increases the availability of clear-sky data in wet tropical regions (Maeda et al., 2016). Here, we found that after correction with the MAIAC algorithm, EVI data from the evergreen forest region do not present any significant difference between the two rainy seasons (Fig. 5b), and basin-wide MAIAC EVI appears to peak during SON rather than MAM (Fig. 4a). Furthermore, direct estimation of photosynthetic rates using SIF/PAR data reveals greater productivity in SON than MAM throughout the evergreen forest band (Fig. 5b) and the basin as a whole (Fig. 4b). The misaligned seasonal peaks of SIF/PAR and $ET_{wb}$ indicate

that either a) water use efficiency (WUE) varies seasonally when generalized across the basin, resulting in higher MAM transpiration but lower MAM photosynthesis, or b) that the MAM peak in ET is primarily driven by direct evaporation from the canopy or land surface rather than transpiration.

### 4.2.1 Leaf age-related WUE variations

   Leaf age offers a possible explanation for the variable WUE hypothesis. Studies of tropical trees have found that new

leaves can take 1–2 months to reach peak photosynthetic capacity and WUE, and that both traits tend to decline as leaves reach 5–6 months old (Sobrado, 1994; Shirke, 2001). Leaf age has been linked to photosynthetic seasonality within the Amazon rainforest (Wu et al., 2016), although this effect has not been investigated in the Congo basin. Connecting the phenology of Congolese forests to basin-wide photosynthesis and transpiration must account for multiple broad ecoregions that span the





equator and therefore face inverted seasonalities (Figs. 5, S2). In much of the deciduous woodlands of the northern (southern)
basin, vegetation leaf flushing tends to begin 1–2 months prior to the onset of MAM (SON) rains and senescence begins around
the end of the SON (MAM) rains, but both processes occur over the course of 1–2 months (Guan et al., 2014; Vinya et al.,
2019). Likewise, microwave backscatter data from large areas of the Congo's evergreen forest imply dry season leaf flushing
likely occurs during DJF and JJA (Guan et al., 2013; Konings et al., 2017). Note, however, that phenological synchronicity
appears to be low in the evergreen forests of central Africa (Couralet et al., 2013), so the magnitude and effects of evergreen
leaf flushing are probably smaller and more temporally distributed in the evergreen forests than in the deciduous woodlands
of the basin.

After accounting for the different ecoregions within the Congo Basin, leaf age effects alone appear unable to explain
the flipped seasonality of $ET_{wb}$ in the two rainy seasons: although WUE increases as leaves mature, overall transpiration rates
in tropical deciduous leaves remain high until they reach old age (Sobrado, 1994; Shirke, 2001). Thus, July–August leaf
flushing of the southern deciduous woodlands, which far exceed the northern woodlands in area (Fig. S2), would most likely
increase basin-wide transpiration alongside photosynthesis during SON. While our current understanding of regional
phenology appears broadly consistent with remotely-sensed vegetation data, better field observations of phenology, leaf age,
and associated changes in stomatal conductance and productivity in the different ecosystems of the Congo basin are needed to
fully determine the role of vegetation in modulating ET seasonality.

**4.2.2 VPD and temperature**

Climatic conditions beyond precipitation could also contribute to the seasonal variations in WUE. For instance, high
VPD reduces WUE by drawing more water from stomata per unit of carbon intake during transpiration. Basin-averaged VPD
data from the ERA-Interim reanalysis do not indicate that MAM conditions are significantly less humid than SON conditions
(Fig. 4e), suggesting seasonal VPD variations cannot explain the flipped $ET_{wb}$–precipitation wet season magnitudes. But the
uncertainties in reanalysis-based temperature and humidity data from tropical regions (Lorenz and Kunstmann, 2012; Brands
et al., 2013) warrant an examination of the limited in-situ data available from within the Congo Basin. Observational VPD
data from weather reports show that while ERA-Interim captures the shapes of seasonal VPD cycles fairly accurately, it fails
to capture the magnitude of daytime VPDs in the evergreen rainforest (Fig. S1). Further examination of station and reanalysis
data from within the evergreen forest region shows that rainforest VPD is slightly greater during MAM than SON (Figs. 5b,
S1c–f), consistent with models based on historical pan evaporation data across the basin (Bultot, 1971) and historical
atmospheric humidity data from within the equatorial rainforest (Lauer, 1989). However, because evergreen rainforest VPDs
are generally low compared to other regions of the basin (Fig. 5) and are only slightly greater during MAM (Fig. S1c–f), VPD
is not expected to be a significant driver of ET variations at basin-wide scales.

While basin-averaged $T_a$ does not vary drastically throughout the year, $T_s$ features a bimodal seasonality that peaks
primarily in September and also from February to March (Fig. S3). $T_s$ can regulate ET via stomatal conductance, which tends
to increase with leaf temperature to a point, although a wide range of sometimes-contradictory results have been published on





this matter (Urban et al., 2017). In this case, the poor alignment of peak $T_s$ and $ET_{wb}$ values—taken in conjunction with the widespread stomatal closures known to occur in tropical forests during the hottest parts of the day (Fisher et al., 2006; Konings and Gentine, 2017; Konings et al., 2017)—indicate that other variables are more directly responsible for the high $ET_{wb}$ observed

during MAM.

### 4.2.3 Radiative fluxes

The magnitude and quality of radiative fluxes can drive the dynamics of $ET_{wb}$ by influencing primary production as well as WUE. $R_n$ and total PAR are both diminished in SON relative to MAM levels (Fig. 4c), which does not explain the SON peak in primary production observed in SIF/PAR and EVI data (Fig. 4a, b). However, greater $R_n$ levels during MAM

could decrease WUE by increasing water demand, thereby driving the high $ET_{wb}$ levels observed in MAM.

Additionally, the quality of PAR beyond the flux's magnitude could affect WUE and explain the apparent decoupling of productivity from irradiance levels and $ET_{wb}$. Prior studies have found that increasing $PAR_{diff}/PAR$ can increase WUE and light use efficiency (LUE) in tropical savannahs and global forest canopies, including in an Amazonian tropical broadleaf stand (Alton et al., 2007; Kanniah et al., 2013). Furthermore, total canopy ET has been found to decrease as the diffuse light fraction

increases (Rocha et al., 2004). Philippon et al. (2019) examine diffuse and direct irradiance data from the Breathing Earth System Simulator (BESS; Ryu et al., 2018) and the Satellite Application Facility for Climate Monitoring (CM-SAF; Müller et al., 2015) and find that the ratio of direct irradiance is often higher during MAM than in SON throughout much of the Congo Basin. The CERES data produce similar results: while $PAR_{diff}/PAR$ peaks in May, on average it is significantly lower during the MAM season than during SON. $PAR_{diff}/PAR$ remains low throughout March (the month when $ET_{wb}$ peaks) and the greater

total PAR flux in MAM is mostly attributable to greater $PAR_{dir}$ (Fig. S4). After removing seasonal cycles and long-term trends, monthly mean SIF/PAR displays a strong negative correlation with mean $PAR_{dir}$ but not with $PAR_{diff}$, indicating photosynthetic rates do not scale as well with increasing direct sunlight as with diffuse sunlight (Fig. S5). These experiments suggest the quality of insolation during SON could allow for higher photosynthetic rates with lower ET than during MAM, especially since monthly $PAR_{diff}/PAR$ in the Congo remains below 0.4 on average (Kanniah et al., 2013) and lower $R_n$ corresponds to

lower water demand (Philippon et al., 2019). Even if total PAR availability would favor a productivity peak in MAM, lower WUE and LUE could result in plants transpiring all available water without reaching the productivity levels achieved in SON. Thus insolation quality potentially explains $ET_{wb}$'s imbalanced seasonality.

### 4.2.4 Terrestrial water storage

The availability of water in the rooting zone can modulate $ET_{wb}$ by directly limiting transpiration during SON. Even

though $P$ during MAM is significant, $dS/dt$ (which measures groundwater within the rooting zone as well as in deeper reserves) remains much lower than its SON levels and only seems to recharge $S$ enough to compensate for the slightly negative $dS/dt$ values of January and February (Fig. 2), indicating that water reserves are largely saturated during MAM and undersaturated at the onset of SON (Fig. 4f). This hypothesis is consistent with prior soil moisture modeling efforts, which indicate that





Congolese ecosystems south of the equator (i.e. most of the basin's area) feature low soil moisture as SON rains begin before
maintaining high soil moisture and deeper water reserves through the end of the MAM rainy season (Pokam et al., 2012; Guan
et al., 2014). Terrestrial water reserves are well-known to modulate productivity/ET throughout the basin (Saeed et al., 2013;
Guan et al., 2014; Ndehedehe et al., 2018; Cuthbert et al., 2019), so depleted $S$ during September and October could limit
transpiration even after the dry season ends. This hypothesis is also consistent with recent findings that variability in soil
moisture and groundwater are much more influential than variability in rainfall with regard to productivity anomalies within
the Congo Basin (Madani et al., 2020), making variations in plant-available soil moisture a plausible driver of the seasonal
$ET_{wb}$ imbalance when combined with the light quality impacts on WUE outlined in the previous section.

### 4.2.5 Direct canopy and soil evaporation

The WUE hypotheses explored above largely focus on the seasonality of transpiration, but seasonal variability in
direct evaporation from forest canopies and the soil system could also potentially influence the seasonal cycle of $ET_{wb}$. As
outlined above, $R_n$ is lower in SON than during MAM (Fig. 4c), suggesting that there may simply be less energy driving direct
evaporation of water from the land surface. As $R_n$ and $ET_{wb}$ both peak in March and high water content in the soil surface layer
is apparently sustained throughout the rainy seasons (Guan et al., 2014), the conditions seem appropriate to drive increased
direct evaporation during MAM. However, the higher proportion of nighttime rains during these months (Philippon et al.,
2016) make this scenario less plausible, as recent rainfall would have more time to drip from the canopy to the soil surface and
percolate to deeper soil levels. Additionally, a recent study of several climate models within the Congo Basin determined that
soil and canopy evaporation rates are likely similar between the two wet seasons, and that the $ET_{wb}$ seasonal imbalance is more
likely due to increased transpiration during MAM (Crowhurst et al., 2020). More research is required to definitively
characterize the role of direct evaporation in the seasonality of ET, but given the dominance of transpiration as the primary
component of ET within the Congo Basin (Lian et al., 2018) and the conclusions of Crowhurst et al. (2020), we find it unlikely
to be the main driver of $ET_{wb}$'s MAM peak.

Taken together, the analyses above suggest that the timing of peak ET rates in MAM rather than SON is primarily
due to a combination of greater moisture availability and higher $R_n$ during MAM, as well as the higher fraction of diffuse
radiation during that season. The decoupled seasonal peaks of $ET_{wb}$ and photosynthetic productivity also indicate that seasonal
variations in WUE occur across the Congo Basin.

### 4.3 Comparison to global ET products

The six global ET models evaluated generally do not capture the amplitudes of interannual and seasonal variability displayed
by $ET_{wb}$ (Fig. 3, Table 3). Broadly speaking, FLUXCOM and to a lesser degree GLDAS-Noah appear to lead global ET
products in reproducing $ET_{wb}$. However, both of these products have significantly lower temporal standard deviations than
$ET_{wb}$ does. Indeed, all alternative ET products have less temporal variability than $ET_{wb}$, ranging from only ~40% of the $ET_{wb}$
variability (FLUXNET-MTE) to at most 80% of it (MERRA-2). These metrics reflect the fact that global ET models tend to





repeat the same seasonal ET cycle every year with only minor interannual variability. Although some products feature relatively accurate seasonal cycles (Fig. 3), $ET_{wb}$ features significant departures from its mean season cycle in any given year (Fig. 1) that are not reflected in the six ET products evaluated here.

Our results show some disagreements with the previous model comparison efforts of Liu et al. (2016). As part of a
global water balance-based assessment of ET products, Liu et al. (2016) also compare $ET_{wb}$ (estimated using a different approach; see below) to ET from GLEAM, FLUXNET-MTE, GLDAS-NOAH Version 2, and MERRA in the Congo Basin. However, in our calculation, the Taylor skill score of GLDAS-2.1 appears to far outperform its value in Liu et al. (2016), FLUXNET-MTE and MERRA2 both show modest improvements in their Taylor scores, and GLEAM v3.1a falls slightly short of its previous value. These discrepancies were likely caused by a) the different study period used by Liu et al. (2016), which
necessitated an extrapolation of GRACE data prior to 2002, b) the exclusive use of the GPCC Version 6 product to constrain $P$ in their water balance models, and c) the use of different versions of the four datasets common to both Liu et al. (2016) and the present study. Nonetheless, comparison of these Taylor scores indicates that the most recent generation of ET products generally improved upon the previous generation (GLEAM being the only partial exception): FLUXCOM outperforms its predecessor, FLUXNET-MTE; GLDAS-NOAH Version 2.1 outperforms Version 2.0; and MERRA Version 2 improves upon
MERRA Version 1's skill score.

Out of the six comparison products, the two remote sensing-based products diverge the most from $ET_{wb}$, despite being the most observationally-driven. While some previous studies have found MODIS-derived ET to approximate actual ET fairly well in nearby regions in West Africa (Schüttemeyer et al., 2007; Opoku-Duah et al., 2008; Andam-Akorful et al., 2015), the results of this comparison reinforce the value of $ET_{wb}$ as a data-driven, independent estimate of a critical hydrological flux.

**4.4 Effects of long-term climatic shifts on ET**

From 2002–2016, no significant trend is detectable in the deseasonalized $ET_{wb}$ data, nor in any of the other water balance component fluxes (Fig. 6). The lack of significant trends in water balance components over the fifteen-year study period is surprising given the numerous reports of declining precipitation in the Congo Basin, both in magnitude (Asefi-Najafabady and Saatchi, 2013; Diem et al., 2014; Zhou et al., 2014; Hua et al., 2016; Dezfuli, 2017) and seasonality (Jiang et al., 2019).
However, the absence of a trend in $P_{TC}$ does not indicate the absence of a longer-term drying trend that began in the 20th century—rather, it probably results from our study period, which is shorter and generally more recent than those of the aforementioned studies, and our analysis of rainfall over all seasons rather than during certain three-month windows. Indeed, careful examination of long-term precipitation plots reveals marked declines in rainfall during the 1990s and early 2000s that did not continue significantly into our 2002–2016 study period (Diem et al., 2014; Dezfuli, 2017; Hua et al., 2019). The lack
of an interannual $ET_{wb}$ trend is consistent with the recent findings of Weerasinghe et al. (2020).

The lack of long-term trends in $ET_{wb}$ and EVI is nevertheless surprising given the changes detectable in the other environmental variables (Figs. 7, S6): $PAR_{diff}$, $PAR_{dir}$, $R_n$, and VPD all increase significantly from 2002–2016, indicating the Congo Basin has become sunnier and less humid in recent years. While meteorological data are quite sparse in the basin during





this fifteen-year period, the increasing VPD trends in ERA-Interim are consistent with projections of rising VPD in CMIP5 models (Yuan et al., 2019). Sunnier and less humid conditions would typically lead to lower WUE in plants, which would in turn lead to increased ET and/or decreased EVI. Given the seasonal dependence of productivity and transpiration on irradiance levels (see Sect. 4.2.3), the apparent lack of a corresponding long-term relationship indicates that some mechanism may be counteracting the biological impacts of rising irradiance and VPD.

Carbon fertilization offers one possible explanation for the lack of $ET_{wb}$ and EVI trends. Rising VPD can indeed
reduce the WUE of vegetation, but conversely, rising atmospheric $CO_2$ levels can increase WUE in tropical forests by catalyzing stomatal closure (De Kauwe et al., 2013; Keenan et al., 2013; Van Der Sleen et al., 2015). Ukkola and Prentice (2013)'s vegetation dynamics simulations yield a sizeable decrease in stomatal conductance between 1960 and 2000 in the Congo Basin, consistent with altered stomatal behavior from $CO_2$ fertilization and increasing VPD. Increasing PAR levels also could have helped support forest productivity rates as stomatal conductance declined, but the PAR data in Fig. S6 suggest a
continual decrease in diffuse PAR fraction that could adversely affect the WUE and LUE of Congolese forests (Kanniah et al., 2013). In summary, even as VPD and irradiance have increased and driven up evaporative demand in plant stomata, the rising concentration of atmospheric $CO_2$ has seemingly allowed the Congo's forests to lower stomatal conductance without significantly impacting growth or ET (Peñuelas et al., 2011; Van Der Sleen et al., 2015). But the effects of present and future carbon fertilization on WUE remain highly uncertain (Guerrieri et al., 2019). Thus, although ET did not show any statistically-
significant trends during our 2002–2015 study period, future ET rates could nevertheless start declining if the compensation between decreasing radiation quality and rising VPD on the one hand and increasing $CO_2$ on the other hand becomes imbalanced, or if long-term declines in precipitation continue.

## 5 Conclusions

This study leverages several remotely-sensed and gauge-based precipitation datasets, river gauge data, and terrestrial water
storage anomalies from GRACE to produce ET estimates for the Congo Basin in central Africa. This technique has been successfully applied to the Amazon Basin in recent years, but to the authors' knowledge it has not yet been used in the Congo Basin except as part of global-scale reviews of major river basin. The Congo Basin is greatly understudied despite its importance as one of the world's largest river basins and one of three major humid tropical forest regions, and quantification of basin-wide ET and its variability is imperative for understanding the basin's influence on regional and global climates as
well as its susceptibility to environmental disturbances. We find annual $ET_{wb}$ to equal 117.2±3.5 cm/year, on average, from 2003–2015—well in line with many historical estimates of basin-wide ET.

Triple collocation was applied to determine the accuracy of $P$ products over the sparsely-gauged Congo Basin, finding that the recently-developed NIC131-gridded dataset is the most accurate over our study period (RMSE of 0.65 cm/month). NIC131-gridded is followed by CHIRPS2 in terms of accuracy (RMSE of 0.93 cm/month), while three separate products that
incorporate GPCC rain gauge data feature similar variability and RMSEs (1.60 to 1.67 cm/month). RMSEs from TC were also



used to create a unified $P_{TC}$ time series with a mean annual precipitation of 150.4±2.6 cm/year. A suite of global ET products is also evaluated versus $ET_{wb}$, with FLUXCOM and GLDAS-Noah Version 2.1 displaying the closest agreement with $ET_{wb}$ from 2002–2011. However, all ET models underestimated the seasonal and interannual variability of $ET_{wb}$.

585       In good agreement with existing literature, rainfall appears to exert a primary control on ET, but other environmental drivers appear to modulate ET and cause unexpected seasonal features, such as the MAM peak in ET recently explored by Crowhurst et al. (2020). Several possible causes for this MAM ET peak were investigated, but neither VPD, temperature, phenology, or leaf age seasonalities could explain this MAM peak. Instead, the amount and quality of radiative energy and the availability of water in the terrestrial system appear to offer the most plausible explanation for the seasonal imbalance in peak $ET_{wb}$—higher diffuse PAR fractions and lower $R_n$ during SON allow for higher WUE, while depleted terrestrial water stores

limit the amount of water available for transpiration. On interannual timescales, VPD, $R_n$, and both direct and diffuse PAR increased from 2002–2016 while no trend was detectable in EVI and $ET_{wb}$, implying the rising concentration of atmospheric $CO_2$ has compensated for the increasingly dry conditions facing the Congo Basin's forests. However, these effects may not remain balanced in a future of higher $CO_2$ levels, increased VPD and temperatures, and spreading deforestation within the basin.

**Author contributions**

AGK and MWB designed the study. MWB and GRQ retrieved the data. All authors analyzed the data. MWB prepared the manuscript with the assistance of AGK and GRQ.

**Data availability**

We provide our monthly basin-wide $P$, $ET_{wb}$, $dS/dt$, discharge, and other data online at
http://doi.org/10.17605/OSF.IO/JPVMB.

Most of the original data products used in this study are freely available to the public: the HydroSHEDS basin boundary shapefile can be retrieved from https://hydrosheds.org/page/hydrobasins. PERSIANN-CDR data were accessed at http://dx.doi.org/10.7289/V51V5BWQ and CHIRPS2 data were downloaded from https://www.chc.ucsb.edu/data. The GPCC
dataset was accessed at https://www.esrl.noaa.gov/psd/. GRACE data were retrieved from GRCTellus Land at http://grace.jpl.nasa.gov, and Congo River discharge data were downloaded from HYBAM at http://www.ore-hybam.org. FLUXNET-MTE and FLUXCOM were made available by the Max Planck Institute for Biogeochemistry; FLUXCOM may be accessed at http://fluxcom.org. GLEAM data were accessed at http://gleam.eu. TRMM, MERRA-2, and GLDAS-Noah data were provided by NASA GES DISC at https://disc.gsfc.nasa.gov/. CERES data were downloaded from
https://ceres.larc.nasa.gov/order_data.php. ERA-Interim data were retrieved from the ECMWF at



https://www.ecmwf.int/en/forecasts/datasets. MOD16A2 and MCD12C1 data were downloaded from the USGS website at https://lpdaac.usgs.gov/products/mod16a2v006/ and https://lpdaac.usgs.gov/products/mcd12c1v006/, respectively, and AIRS data were downloaded from NASA's Giovanni web interface (https://giovanni.gsfc.nasa.gov/giovanni/). ASOS and MIDAS weather data were sourced from the UK Met Office's CEDA website at

http://catalogue.ceda.ac.uk/uuid/220a65615218d5c9cc9e4785a3234bd0 and from Iowa State University's Iowa Environmental Mesonet site at https://mesonet.agron.iastate.edu/ASOS/.

**Competing interests**

The authors declare that they have no conflict of interest.

**Acknowledgements**

We gratefully acknowledge the researchers who provided their datasets and expertise for this study: A. Lyapustin and Y. Wang of NASA provided the MAIAC EVI data; S. Nicholson and D. Klotter of Florida State University provided their NIC131-gridded dataset; and J. Joiner of NASA provided the GOME-2 SIF data. C. Frankenberg of the California Institute of Technology graciously provided additional SIF data for comparison. MWB was supported by a Stanford University Vice Provost for Undergraduate Education (VPUE) grant. This work was also supported by NASA Terrestrial Ecology award

80NSSC18K0715 through the New Investigator Program, by the NASA Carbon Cycle Science program, and by NOAA grant NA17OAR4310127.

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
