# Peer review of "Data-driven estimates of evapotranspiration and its controls in the Congo Basin"

_Hydrology and Earth System Sciences, 2020_

## Referee Comment (RC1) · Anonymous Referee #1 · 20 May 2020

Review of Data-driven estimates of evapotranspiration and its drivers in the Congo Basin by Burnett et al.,

General Comments:

This study produces a monthly basin-wide ET estimates for the Congo basin based on the water balance approach for the period April 2002 to December 2016 evaluated against previous literature estimates and six global ET products. Drivers were investigated for the seasonal and interannual variability. This study is very well structured and written, with a good flow and ease to read and understand the concepts being used. The author has been thorough in his review of existing studies, making good comparisons and analyses to their findings throughout the paper. I have a few minor comments as follows.

It would be interesting if you could discuss whether the use of this technique can be easily applied to other basins even without the use of a precipitation product being developed specifically for a particular basin under study? The findings are very interesting and can be useful for other larger basins in Africa such as the Niger basin with a large delta and the Nile basin.

You compare the ETwb with six available global ET products. As I see you have referenced Weerasingle et al., that also found that GLEAM and MOD16A2 were substantially underestimating ET at the long-term annual scale, whereas other products within that study had much lower biases. It would be interesting to see the same comparison with one or more of the products with lower biases in their study to see if those products capture the seasonal variations and bias better than the products being analysed in this study.

Specific Comments:

P3L65 – you say that studies assume no change in water storage when applying the water balance equation and state 'an assumption with little support'. Is this really true? In the study previously mentioned, Weerasinghe et al., they have looked at the largest contribution of the change in annual water storage from a study using GRACE data and found this to be 20mm yr-1. They then applied this to several basins in Africa including the Congo basin and found there to be only a 2.3% representation of total long-term annual average ET. I believe this assumption has been made for large watersheds and for long-term averages and there is actually a lot of existing studies to support this. Be careful with this statement. Also, In all studies, if it is possible to use the change in storage, this should always give better results, although it may be negligible depending on the timescale and size of area and thus may not be the most important aspect of their studies if not looking at smaller temporal and spatial scales.

Technical Comments:

P1L10 – '…second-largest river basin *in* the world…'

Table 1 caption has 'in' twice consecutively.

P14L355 – I would mention here the three variables you are considering in the text so the reader does not have to go to the figure before they know.

---

## Referee Comment (RC2) · Anonymous Referee #2 · 19 Jun 2020

Review of Data-driven estimates of evapotranspiration and its drivers in the Congo Basin By Burnett et al

This article presents (1) the development of evapotranspiration estimate for the whole of the Congo Basin based on a water balance equation where components are either derived from up-to-date satellitate estimates (P, ds/dt) or in-situ measurements (Q, P), (2) a comparison with 6 existing products developed at global scale from different sources (model, reanalyses, satellites, in-situ) and (3) an analysis of the drivers of ET variations at the mean annual and interannual scale.

I find the article particularly well written and structured, with a carefull use of the multiple databases selected for analysis and interesting discussion sections regarding the drivers. A lot of the papers cited are also quite recent ones so that this paper itself is

a nice review of the last findings for the CB. It is also particularly relevant as the functionning and inter-relationships of the hydrologic, biospheric and climatic components of the Congo Basin are far less well known and documented than those of Amazonia. An interesting finding is the potential role of the radiative forcing.

I do not have any major comment, just a few minor ones listed below to improve somewhat the paperĂă:

L125Ăă: amongst the quite recent and few attempts of rainfall estimates validation against in-situ measurements for CB you could also refer to Camberlin et al 2019 QJRMS

L247Ăă: I would specify here ÂńÂăcorrelation at the interannual time-scaleĂăÂż (and not for the mean annual cycles)

Fig1 + all your figuresĂă: would be very helpful to readers if gridlines were provided so that we can better pick the peaks and lows, the lead-lags between variables etc

L286Ăă: part of the discussion about the shape of the mean annual cycle with a minima in JJA being driven by the southern pixels (which are dominant in the CB, e.g. L360) should appear as soon as this section

L289Ăă: to me when reading the figure the fastest S decrease is between April and May . . .

L308-311Ăă: these differences in terms of dynamics, amplitude and ratio between MAM and SON are an important point that certainly plays then on the analysis of drivers . . . your ET estimate is singular from this point of view. you might provide these two statistics (amplitude, ratio) to support your discourse . . .

Table 3Ăă: why don't you provide the annual mean on 2002-2011 if you compute correlations etc on this periodĂă??? You should also provide the significance level for the correlations . . .

[Figure]

L346Âă: could you explain a bit the added value of normalising SIF by PARÂă

L352Âă: I dont' think that the ratio is ÂńÂăsignificantlyÂăÂż greater in SON as compared to MAM as the greatest ratio is observed in MayÂă; it seems quite comparable (and easy to check) so to my opinion this does not play ...

L371Âă: the positive trend in radiation might not be a regional signal only as many studies have shown a brightening at global scale the past decade (as opposed to a dimming in the 90s)

L400Âă: the scale of your study is far larger than the ones by Betbeder et al and Philippon et al which focus on very specific forests growing on particular soils, therefore I would not be so affirmative about a unique and consistant peak of EVI in SON across the whole central Africa forests even if these authors have not used the last state of the art EVI product...

4.2.4 water storageÂă: Your discussion is valid for southern pixels but not the northern ones ... while the wettest rainy season is SON over the whole CB, the driest rainy season is DJF to the north and JJA to the south so this changes the dynamics of water storage and available water for trees at the beginnign of each rainy season ...

L540Âă: I would also add, amongst the reasons why you do not capture the negative trend in rainfall, that this trend mainly affects the northern part of the CB and your index is mainly ÂńÂădrivenÂăÂż by southern pixels

Fig.2SÂă: would be good that the CB be contextualised ie by presenting a larger map of Central Africa with countries borders ....

Lastly I would have liked seeing in this paper a short ÂńÂăperspectiveÂăÂż section on the further analyses these results call for and the study limits (unfortunately ET at the monthly and CB scale that does not allow documenting spatial variations in the drivers of ET that might be significant nor fine temporal - daily or infra-daily – variations which might be key to understand differences between the two wet or the two dry periods)

Please also note the supplement to this comment:
https://www.hydrol-earth-syst-sci-discuss.net/hess-2020-186/hess-2020-186-RC2-supplement.pdf

---

## Author Comment (AC1) · 1 Jul 2020

**Author response to Referee Comment 1 on Burnett et al. (HESS-2020-186):**

**Referee Comment:** This study produces a monthly basin-wide ET estimates for the Congo basin based on the water balance approach for the period April 2002 to December 2016 evaluated against previous literature estimates and six global ET products. Drivers were investigated for the seasonal and interannual variability. This study is very well structured and written, with a good flow and ease to read and understand the concepts being used. The author has been thorough in his review of existing studies, making good comparisons and analyses to their findings throughout the paper. I have a few minor comments as follows.

**Author Response:** We thank the referee for their positive view of the article.

**Referee Comment:** It would be interesting if you could discuss whether the use of this technique can be easily applied to other basins even without the use of a precipitation product being developed specifically for a particular basin under study? The findings are very interesting and can be useful for other larger basins in Africa such as the Niger basin with a large delta and the Nile basin.

**Author Response:** In the revised draft we will include text in discussion section 4.1 regarding this topic.

**Referee Comment:** You compare the ETwb with six available global ET products. As I see you have referenced Weerasingle et al., that also found that GLEAM and MOD16A2 were substantially underestimating ET at the long-term annual scale, whereas other products within that study had much lower biases. It would be interesting to see the same comparison with one or more of the products with lower biases in their study to see if those products capture the seasonal variations and bias better than the products being analysed in this study.

**Author Response:** This is a good idea. In the revised paper we will include SSEBop as a seventh global ET product comparison, as it is the global product recommended by Weerasinghe et al. with the lowest bias over the Congo Basin and the most use in the literature.

**Referee Comment:** P3L65 – you say that studies assume no change in water storage when applying the water balance equation and state 'an assumption with little support'. Is this really true? In the study previously mentioned, Weerasinghe et al., they have looked at the largest contribution of the change in annual water storage from a study using GRACE data and found this to be 20mm yr-1. They then applied this to several basins in Africa including the Congo basin and found there to be only a 2.3% representation of total long-term annual average ET. I believe this assumption has been made for large watersheds and for long-term averages and there is actually a lot of existing studies to support this. Be careful with this statement. Also, In all studies, if it is possible to use the change in storage, this should always give better results, although it may be negligible depending on the timescale and size of area and thus may not be the most important aspect of their studies if not looking at smaller temporal and spatial scales.

**Author Response:** This is a good point. We agree that the language in the previous version of our manuscript may be too strong, although we do maintain the importance of incorporating terrestrial water storage changes in water balance estimates of ET at monthly timescales. As the reviewer points out, many studies of long time periods and large watersheds use the assumption of constant TWS to simplify their methods, and in this context the assumption seems justifiable. However, Congo Basin GRACE data from our study (as well as Crowley et al. 2006 and Rodell et al. 2018) indicate that monthly dS/dt values can exceed 5 cm/month. Given ET values between 5 and 15 cm/mo, TWS is clearly worth incorporating for studies like ours which aim to quantify month-to-month variability in ET. We will revise the manuscript to clarify this point.

**Technical Comments:**

**Referee Comment:** P1L10 – '…second-largest river basin *in* the world…'

**Author Response:** Corrected in revised manuscript.

**Referee Comment**: Table 1 caption has 'in' twice consecutively.

**Author Response:** Corrected in revised manuscript.

**Referee Comment:** P14L355 – I would mention here the three variables you are considering in the text so the reader does not have to go to the figure before they know.

**Author Response:** The clarification on P14L355 has been added to the text. We thank the referee for their attentive reading.

**References:**

Crowley, J. W., Mitrovica, J. X., Bailey, R. C., Tamisiea, M. E. and Davis, J. L.: Land water storage within the Congo Basin inferred from GRACE satellite gravity data, Geophys. Res. Lett., 33, L19402, https://doi.org/10.1029/2006GL027070, 2006.

Rodell, M., Famiglietti, J. S., Wiese, D. N., Reager, J. T., Beaudoing, H. K., Landerer, F. W. and Lo, M. H.: Emerging trends in global freshwater availability, Nature, 557, 651–659, https://doi.org/10.1038/s41586-018-0123-1, 2018.

Weerasinghe, I., Bastiaanssen, W., Mul, M., Jia, L. and van Griensven, A.: Can we trust remote sensing evapotranspiration products over Africa?, Hydrol. Earth Syst. Sci., 24, 1565–1586, https://doi.org/10.5194/hess-24-1565-2020, 2020.

---

## Author Comment (AC2) · 1 Jul 2020

**Author Response to Referee Comment 2 on Burnett et al. (HESS-2020-186):**

**Referee Comment:** This article presents (1) the development of evapotranspiration estimate for the whole of the Congo Basin based on a water balance equation where components are either derived from up-to-date satellitate estimates (P, ds/dt) or in-situ measurements (Q, P), (2) a comparison with 6 existing products developed at global scale from different sources (model, reanalyses, satellites, in-situ) and (3) an analysis of the drivers of ET variations at the mean annual and interannual scale.

I find the article particularly well written and structured, with a carefull use of the multiple databases selected for analysis and interesting discussion sections regarding the drivers. A lot of the papers cited are also quite recent ones so that this paper itself is a nice review of the last findings for the CB. It is also particularly relevant as the functionning and inter-relationships of the hydrologic, biospheric and climatic components of the Congo Basin are far less well known and documented than those of Amazonia. An interesting finding is the potential role of the radiative forcing.

**Author Response:** We thank the referee for their positive view of the article.

**Referee Comment:** L125 : amongst the quite recent and few attempts of rainfall estimates validation against in-situ measurements for CB you could also refer to Camberlin et al 2019 QJRMS

**Author Response:** Thank you for pointing us to this study; we will add it to the article.

**Referee Comment:** L247 : I would specify here « correlation at the interannual time-scale » (and not for the mean annual cycles)

**Author Response:** We will make this clarification in the revised article.

**Referee Comment:** Fig1 + all your figures : would be very helpful to readers if gridlines were provided so that we can better pick the peaks and lows, the lead-lags between variables etc

**Author Response:** While gridlines were omitted to preserve the clarity of the figures, we recognize that the subject matter of this paper will lead readers to search for specific minima and maxima in the many plots included, and as such gridlines would be useful in many figures. We are planning to add gridlines to several of our figures in the revised article, except where multiple vertical axes are present.

**Referee Comment:** L286 : part of the discussion about the shape of the mean annual cycle with a minima in JJA being driven by the southern pixels (which are dominant in the CB, e.g. L360) should appear as soon as this section

**Author Response:** We will add language to Results section 3.2 to introduce this point earlier.

**Referee Comment:** L289 : to me when reading the figure the fastest S decrease is between April and May …

**Author Response:** Note that Figure 2 plots *dS/dt* rather than *S*; we will edit the corresponding paragraph for clarity. Figure 4f shows the basin-wide *S* annual cycle.

**Referee Comment:** L308-311 : these differences in terms of dynamics, amplitude and ratio between MAM and SON are an important point that certainly plays then on the analysis of drivers … your ET estimate is singular from this point of view. you might provide these two statistics (amplitude, ratio) to support your discourse …

**Author Response:** Thanks for the suggestion. We will include these statistics in Table 3 to support comparison of the different ET products.

**Referee Comment:** Table 3 : why don't you provide the annual mean on 2002-2011 if you compute correlations etc on this period ??? You should also provide the significance level for the correlations …

**Author Response:** $ET_{wb}$ data begin in April 2002, so 2002 was not included in the mean annual ET calculations, whereas all available months were used to generate the rest of the statistics in Table 3. We initially made this choice to avoid calculating an annual mean over a time period that did not represent an integer number of years. However, we recognize that this was confusing to readers. In the revised manuscript we are planning to incorporate (at the suggestion of Reviewer 1) a seventh global ET product (SSEBop) that is only available from 2003 onward, so consequently all statistics will be recalculated for 2003-2011.

**Referee Comment:** L346 : could you explain a bit the added value of normalising SIF by PAR

**Author Response:** Normalizing SIF by PAR is intended to isolate the contributions of climatic and ecophysiological factors to productivity while controlling for the influence of radiation on productivity (see Pagán et al. 2019 and Madani et al. 2017 in *Remote Sensing* for further explanation). This was designed to allow us to read clearer information about moisture availability, light use efficiency, and vegetation dynamics from the SIF data.

The relatively low variance of total PAR means that the SIF seasonal cycle and peak during October/November are similar with and without PAR normalization (see GOME2/CERES plots below, which we will add to the Supplement). We will add some text to the manuscript clarifying these points, and discussing the implications of our PAR-normalized SIF data for light use efficiency and the potential limiters of transpiration.

[Figure]

**Referee Comment:** L352 : I dont' think that the ratio is « significantly » greater in SON as compared to MAM as the greatest ratio is observed in May ; it seems quite comparable (and easy to check) so to my opinion this does not play ...

**Author Response:** You are right that in terms of seasonal means they are not much different. In the revised draft we will clarify that March and April have low diffuse PAR ratios that could cause the high ET observed in those months (in May, when diffuse PAR ratio peaks, ET decreases to nearly SON levels).

**Referee Comment:** L371 : the positive trend in radiation might not be a regional signal only as many studies have shown a brightening at global scale the past decade (as opposed to a dimming in the 90s)

**Author Response:** Dimming/brightening trends in central Africa over our study period are generally not well-constrained to our knowledge, and we had difficulty finding literature on global dimming/brightening for our study period (the recent studies we examined generally neglected Africa due to a lack of measurements). We will add a brief discussion of changing irradiance in central Africa with references to recent literature, including a 2020 assessment of global dimming/brightening trends (Hatzianastassiou et al., 2020), in Section 4.4.

**Referee Comment:** L400 : the scale of your study is far larger than the ones by Betbeder et al and Philippon et al which focus on very specific forests growing on particular soils, therefore I would not be so affirmative about a unique and consistant peak of EVI in SON across the whole central Africa forests even if these authors have not used the last state of the art EVI product...

**Author Response:** This is a good point—we will add text to this section noting the smaller extent of the Betbeder and Philippon studies as a likely cause of the different EVI cycles, and to clarify that the different EVI algorithms used are merely an additional factor on top of the spatial differences.

**Referee Comment:** 4.2.4 water storage : Your discussion is valid for southern pixels but not the northern ones … while the wettest rainy season is SON over the whole CB, the driest rainy season is DJF to the north and JJA to the south so this changes the dynamics of water storage and available water for trees at the beginnign of each rainy season ...

**Author Response:** This is true—we will add text to section 4.2.4 to clarify that much of the discussion pertains to the southern portion of the basin, but that the hydrologic characteristics of the southern region dominate the basin-wide hydrologic cycles and are therefore still relevant to the study.

**Referee Comment:** L540 : I would also add, amongst the reasons why you do not capture the negative trend in rainfall, that this trend mainly affects the northern part of the CB and your index is mainly « driven » by southern pixels

**Author Response:** Good point. We will add text mentioning this to Section 4.4.

**Referee Comment:** Fig.2S : would be good that the CB be contextualised ie by presenting a larger map of Central Africa with countries borders ....

**Author Response:** We will add such a map as a new panel in Figure S2.

**Referee Comment:** Lastly I would have liked seeing in this paper a short « perspective » section on the further analyses these results call for and the study limits (unfortunately ET at the monthly and CB scale that does not allow documenting spatial variations in the drivers of ET that might be significant nor fine temporal - daily or infra-daily – variations which might be key to understand differences between the two wet or the two dry periods)

**Author Response:** We will add a brief section to the end of the Discussion outlining some further analyses that would be useful, including more in-situ observational studies and meteorological stations, continuation of this study using data from the GRACE-FO mission, examination of ET in sub-regions of the Congo Basin, etc.

**References:**

Hatzianastassiou, N., Ioannidis, E., Korras-Carraca, M. B., Gavrouzou, M., Papadimas, C. D., Matsoukas, C., Benas, N., Fotiadi, A., Wild, M., & Vardavas, I.: Global dimming and brightening features during the first decade of the 21st century. Atmosphere-Basel, 11, 308, https://doi.org/10.3390/atmos11030308, 2020.

Madani, N., Kimball, J. S., Jones, L. A., Parazoo, N. C., and Guan, K.: Global analysis of bioclimatic controls on ecosystem productivity using satellite observations of solar-induced chlorophyll fluorescence, Remote Sens., 9, 530, https://doi.org/10.3390/rs9060530, 2017.

Pagán, B. R., Maes, W. H., Gentine, P., Martens, B., and Miralles, D. G.: Exploring the potential of satellite

solar-induced fluorescence to constrain global transpiration estimates, Remote Sens., 11, 413, https://doi.org/10.3390/rs11040413, 2019.

---

## Author Response (AR1)

Dear Dr. Teuling,

Please find attached our revised manuscript, "Data-driven estimates of evapotranspiration and its controls in the Congo Basin" (HESS-2020-186). Thanks to the insightful comments of our two reviewers, we believe the manuscript is much strengthened and will serve as a valuable contribution to the growing body of literature on the hydrology of the Congo River Basin.

While the structure and central narrative of the paper remain essentially unchanged, we have strengthened several components of the article at the suggestion of our reviewers. We now include the energy balance and remote sensing-based SSEBop product into our analysis of global ET products, based on Reviewer 1's suggestion to include global ET products that a recent pan-African study found to be most accurate in the Congo Basin. Reviewer 2 suggested additional metrics for comparing the seasonal cycles of global ET products; we have added these metrics into our Results and Discussion sections as well. Reviewer 2 also prompted us to include more discussion of within-basin spatial variability, previous phenological studies, and PAR normalization of SIF data (for which we have also added an additional figure to the Supplement). Both reviewers wrote that additional perspective on research needs/opportunities would be welcome in the article, and to this end we have added two new paragraphs (one in its own new subsection) to the Discussion section. Numerous other technical details and clarifications have been added throughout the text, and we also slightly reworded the title to sound less redundant.

In addition to the changes brought about by our discussions with reviewers, we made two technical corrections to the article to address errors we uncovered after the initial submission. Comparing our results to recent literature cited by Reviewer 1 led us to realize that our Congo Basin ET data from GLEAM v3.1 and MOD16A2 were erroneously low due to mistakes in processing the data. We have corrected these mistakes and also taken the opportunity to update our data to the more recent GLEAM v3.3 product. While we made significant changes to the text of Sections 3.3 and 4.3 to correctly reflect the performance of the GLEAM and MOD16A2 products, our primary conclusions regarding the failure of global ET models to capture the variability of ET in the Congo remain essentially unchanged. Finally, we noticed that one subplot in the previous version of our manuscript (Fig. 7c) was displaying PAR data without the mean seasonal cycle removed, as was reported in the caption; this error has been fixed and again had no impact on the findings of the article.

We believe we have sufficiently addressed the thoughtful concerns of our reviewers, and that the messages of our article are clearer and more thorough as a result. A point-by-point response to our reviewers' comments is included alongside the marked-up revision of our article.

Sincerely,

Michael W. Burnett

Gregory R. Quetin

Alexandra G. Konings

**Response to Reviews – Burnett et al. (HESS-2020-186)**

**Reviewer 1:**

> *Referee Comment: This study produces a monthly basin-wide ET estimates for the Congo basin based on the water balance approach for the period April 2002 to December 2016 evaluated against previous literature estimates and six global ET products. Drivers were investigated for the seasonal and interannual variability. This study is very well structured and written, with a good flow and ease to read and understand the concepts being used. The author has been thorough in his review of existing studies, making good comparisons and analyses to their findings throughout the paper. I have a few minor comments as follows.*

**Author Response:** We are glad the reviewer took a positive view of the article. Responses to comments are included below.

> *Referee Comment: It would be interesting if you could discuss whether the use of this technique can be easily applied to other basins even without the use of a precipitation product being developed specifically for a particular basin under study? The findings are very interesting and can be useful for other larger basins in Africa such as the Niger basin with a large delta and the Nile basin.*

**Author Response:** In the revised draft we have added the following text in discussion section 4.1 regarding this topic:

"The results of this study also reinforce the value of the inverted water balance method for studying river basins large enough to accommodate the coarse spatial resolution of GRACE data. Compared to the difficulty of directly measuring ET and the large amount of observational data needed to constrain ET models, the inverted water balance is conceptually straightforward and has relatively simple data requirements. But as demonstrated here and in other large river basins like the Amazon (Maeda et al., 2017; Swann and Koven, 2017), inverting the water balance produces robust estimates of ET which can be used to validate and improve other ET models' representation of sparsely-observed basins. Limitations of water balance ET estimates include the coarse spatial resolution, monthly timesteps, and short temporal coverage of GRACE (2002–2016, with various data gaps), the availability of river discharge data for the area of interest, and the quality of gridded $P$ data in the region. However, the use of $dS/dt$ data may not be necessary in long-term ET estimates (Weerasinghe et al. 2020), so the limitations of GRACE data mostly affect studies examining ET variability on annual or shorter timescales. The uncertainties of $P$ datasets can be assessed and mitigated using techniques such as TC (Stoffelen 1998; McColl et al., 2014; Alemohammad et al. 2015; Dong et al. 2020), but basins with more thorough gauge coverage than the Congo probably do not require such detailed analysis of multiple gridded $P$ products."

> *Referee Comment: You compare the ETwb with six available global ET products. As I see you have referenced Weerasingle et al., that also found that GLEAM and MOD16A2 were*

*substantially underestimating ET at the long-term annual scale, whereas other products within that study had much lower biases. It would be interesting to see the same comparison with one or more of the products with lower biases in their study to see if those products capture the seasonal variations and bias better than the products being analysed in this study.*

**Author Response:** In the revised paper we have included SSEBop as a seventh global ET product for comparison, as it is the global product recommended by Weerasinghe et al. with the lowest bias over the Congo Basin and the most use in the literature.

*Referee Comment: P3L65 – you say that studies assume no change in water storage when applying the water balance equation and state 'an assumption with little support'. Is this really true? In the study previously mentioned, Weerasinghe et al., they have looked at the largest contribution of the change in annual water storage from a study using GRACE data and found this to be 20mm yr-1. They then applied this to several basins in Africa including the Congo basin and found there to be only a 2.3% representation of total long-term annual average ET. I believe this assumption has been made for large watersheds and for long-term averages and there is actually a lot of existing studies to support this. Be careful with this statement. Also, In all studies, if it is possible to use the change in storage, this should always give better results, although it may be negligible depending on the timescale and size of area and thus may not be the most important aspect of their studies if not looking at smaller temporal and spatial scales.*

**Author Response:** This is a good point. We agree that the language in the previous version of our manuscript may be too strong, although we do maintain the importance of incorporating terrestrial water storage changes in water balance estimates of ET at monthly timescales. We have revised the paragraph to read as follows:

"While some previous studies have applied a similar technique to the Congo Basin as part of larger-scale experiments, these studies assumed terrestrial water storage was constant over their study periods (Marshall et al., 2012; Ukkola and Prentice, 2013; Weerasinghe et al., 2020)—a plausible assumption for long-term ET estimates, but one that could mask a large degree of ET variability on annual and shorter timescales. Indeed, remotely-sensed evidence suggests water storage anomalies within the basin do change significantly on a monthly and interannual basis (Crowley et al., 2006; Rodell et al., 2018), even if long-term trends are typically small relative to the magnitude of ET fluxes (Weerasinghe et al., 2020). Thus in order to explore seasonal cycles and variations in basin-wide ET, terrestrial water storage must be constrained in inverted water balance models…"

*Referee Comment: P1L10 – '…second-largest river basin in the world…'*

**Author Response:** Corrected in revised manuscript.

*Referee Comment: Table 1 caption has 'in' twice consecutively.*

**Author Response:** Corrected in revised manuscript.

> *Referee Comment:* *P14L355 – I would mention here the three variables you are considering in the text so the reader does not have to go to the figure before they know.*

**Author Response:** The clarification on P14L355 has been added to the text, thank you.

**Reviewer 2:**

> *Referee Comment: This article presents (1) the development of evapotranspiration estimate for the whole of the Congo Basin based on a water balance equation where components are either derived from up-to-date satellitate estimates (P, ds/dt) or in-situ measurements (Q, P), (2) a comparison with 6 existing products developed at global scale from different sources (model, reanalyses, satellites, in-situ) and (3) an analysis of the drivers of ET variations at the mean annual and interannual scale.*

> *I find the article particularly well written and structured, with a carefull use of the multiple databases selected for analysis and interesting discussion sections regarding the drivers. A lot of the papers cited are also quite recent ones so that this paper itself is a nice review of the last findings for the CB. It is also particularly relevant as the functionning and inter-relationships of the hydrologic, biospheric and climatic components of the Congo Basin are far less well known and documented than those of Amazonia. An interesting finding is the potential role of the radiative forcing.*

**Author Response:** We are glad the referee took a positive view of the article.

> *Referee Comment: L125 : amongst the quite recent and few attempts of rainfall estimates validation against in-situ measurements for CB you could also refer to Camberlin et al 2019 QJRMS*

**Author Response:** Camberlin et al. (2019) is now cited several times throughout the study.

> *Referee Comment: L247 : I would specify here « correlation at the interannual time-scale » (and not for the mean annual cycles)*

**Author Response:** This line is clarified in the revised article. The sentence now states: "In order to track interannual correlations between $ET_{wb}$, meteorological variables, and vegetation indices, the Breaks for Additive Season and Trend (BFAST) R package (Verbesselt et al., 2015) was used…"

> *Referee Comment: Fig1 + all your figures : would be very helpful to readers if gridlines were provided so that we can better pick the peaks and lows, the lead-lags between variables etc*

**Author Response:** We have added gridlines to all figures in the paper, although horizontal gridlines are omitted where multiple y-axes are used in single plots, and vertical gridlines are omitted in Figs. 6 and 7 to preserve the visual clarity of the figures. Since the data in these plots will be made available to interested scholars in an online repository after publication (http://doi.org/10.17605/OSF.IO/JPVMB), we believe the gridlines we added will be sufficiently useful for conveying our messages in the text.

*Referee Comment:* L286 : part of the discussion about the shape of the mean annual cycle with a minima in JJA being driven by the southern pixels (which are dominant in the CB, e.g. L360) should appear as soon as this section

**Author Response:** Thank you, this is a good point. This paragraph now begins, "Plotting monthly means of the water balance fluxes provides further clarity regarding their seasonal cycles (Fig. 2). The basin-wide seasonal flux cycles are dominated by contributions from the region south of the equator, which comprises the majority of the Congo Basin (Fig. S2)."

*Referee Comment:* L289 : to me when reading the figure the fastest S decrease is between April and May …

**Author Response:** Fig. 2 plots $dS/dt$ rather than $S$. While $dS/dt$ decreases the fastest between April and May, $S$ does not. To clarify, we have revised this paragraph to include "Positive $dS/dt$ rates indicate $S$ regenerates mostly during the very wet October and November months…".

*Referee Comment:* L308-311 : these differences in terms of dynamics, amplitude and ratio between MAM and SON are an important point that certainly plays then on the analysis of drivers … your ET estimate is singular from this point of view. you might provide these two statistics (amplitude, ratio) to support your discourse …

**Author Response:** We have added these statistics into Table 3 and included discussion of amplitude and ratio in Section 3.3.

*Referee Comment:* Table 3 : why don't you provide the annual mean on 2002-2011 if you compute correlations etc on this period ??? You should also provide the significance level for the correlations …

**Author Response:** We had previously used a different annual mean averaging period to avoid using data-incomplete years (which would bias the annual mean ETs), but we agree that this was confusing. Since we have added a comparison to SSEBop in the revised manuscript (based on Reviewer 1's recommendation), we now evaluate all metrics and averages on the 2003-2011 period anyways, so this issue has been solved.

*Referee Comment:* L346 : could you explain a bit the added value of normalising SIF by PAR

**Author Response:** Normalizing SIF by PAR is intended to isolate the contributions of climatic and ecophysiological factors to productivity while controlling for the influence of radiation on productivity. This was designed to allow us to read clearer information about moisture availability, light use efficiency, and vegetation dynamics from the SIF data. We have added a sentence to Section 2.4 to clarify this point and to reference recent studies that did the same: "SIF was normalized by monthly CERES total PAR data in order to isolate the effects of phenological, physiological, and hydrological

variability on plant productivity independent of radiative controls (Madani et al., 2017; Pagán et al., 2019)."

The relatively low variance of total PAR also means that the SIF seasonal cycle and peak during October/November are similar with and without PAR normalization (see the new Supplemental Figure S3). Text has been added to Section 3.4 to note this: "SIF peaks in October and November with or without PAR normalization (Fig. S3), indicating both greater total photosynthesis and more light-efficient production during these months". In addition, we have further leveraged our use of PAR-normalized SIF in Section 4.2, noting evergreen forest LUE appears to be greater in SON than in MAM.

> *Referee Comment: L352 : I dont' think that the ratio is « significantly » greater in SON as compared to MAM as the greatest ratio is observed in May ; it seems quite comparable (and easy to check) so to my opinion this does not play ...*

**Author Response:** It is true that in terms of seasonal means they are not much different. In the revised line we clarify that March and April have low diffuse PAR ratios that could cause the high ET observed in those months (in May, when diffuse PAR ratio peaks, ET decreases to nearly SON levels):

"WUE during SON may be higher than in March and April (when $ET_{wb}$ peaks) because of the greater ratio of diffuse PAR to total PAR during SON (Fig. 4d) which can increase photosynthetic efficiency (Mercado et al., 2009), as further discussed in Sect. 4.2.3."

> *Referee Comment: L371 : the positive trend in radiation might not be a regional signal only as many studies have shown a brightening at global scale the past decade (as opposed to a dimming in the 90s)*

**Author Response:** This is a good point. We tried to add text related to this point to the manuscript, but dimming/brightening trends in central Africa do not seem well-constrained during our study period in the literature, and we ultimately found there was no succinct way to summarize the relevant point without distracting from the message of the paragraph, which is about how different trends in drivers combine to influence ET trends. Thus, for the sake of clarity of the remaining message, and because it does not affect our interpretation, we have not incorporated it here.

> *Referee Comment: L400 : the scale of your study is far larger than the ones by Betbeder et al and Philippon et al which focus on very specific forests growing on particular soils, therefore I would not be so affirmative about a unique and consistant peak of EVI in SON across the whole central Africa forests even if these authors have not used the last state of the art EVI product...*

**Author Response:** This is a good point—we have added text to this section noting the smaller extent of the Betbeder and Philippon studies as a likely cause of the different EVI cycles, and to clarify that the different EVI algorithms used are merely an additional factor on top of the spatial differences:

"However, these studies focused on specific areas of evergreen forests and wetlands that may not represent the ecohydrology of the Congo Basin's entire equatorial rainforest belt, and used MODIS

products that do not fully account for sun-sensor geometry and other sources of error at low latitudes (Hilker et al., 2012; Bi et al., 2016)."

> *Referee Comment: 4.2.4 water storage : Your discussion is valid for southern pixels but not the northern ones … while the wettest rainy season is SON over the whole CB, the driest rainy season is DJF to the north and JJA to the south so this changes the dynamics of water storage and available water for trees at the beginnign of each rainy season ...*

**Author Response:** This is true—section 4.2.4 has been modified to clarify that much of the discussion pertains to the southern portion of the basin, but that the hydrologic characteristics of the southern region dominate the basin-wide hydrologic cycles and are therefore still relevant to the study:

"As with other variables examined in this study, seasonal $S$ dynamics at the basin scale are driven by the wet SON and dry JJA seasons experienced by the larger southern portion of the basin. Even though basin-wide $P$ during MAM is significant, $dS/dt$ (which measures groundwater within the rooting zone as well as in deeper reserves) remains much lower than its SON levels and only seems to recharge $S$ enough to compensate for the slightly negative $dS/dt$ values of January and February (Fig. 2), indicating that water reserves are largely saturated during MAM and undersaturated at the onset of SON when averaged across the basin (Fig. 4f). This hypothesis is consistent with prior soil moisture modeling efforts, which indicate that Congolese ecosystems south of the equator (i.e. most of the basin's area) feature low soil moisture as SON rains begin before maintaining high soil moisture and deeper water reserves through the end of the MAM rainy season (Pokam et al., 2012; Guan et al., 2014)."

> *Referee Comment: L540 : I would also add, amongst the reasons why you do not capture the negative trend in rainfall, that this trend mainly affects the northern part of the CB and your index is mainly « driven » by southern pixels*

**Author Response:** Thanks for the suggestion - we added the spatial mismatch to the list of reasons presented in this line, along with two references on the northern bias in rainfall declines:

"However, the absence of a trend in $P_{TC}$ does not prove the absence of a longer-term drying trend that began in the 20[th] century—rather, it probably results from our study period, which is shorter and generally more recent than those of the aforementioned studies, our analysis of rainfall over all seasons rather than during certain three-month windows, and the fact that rainfall declines mainly affect the northern portion of the basin (Zhou et al., 2014; Hua et al., 2016) whereas our study is dominated by the basin area south of the equator (Fig. S2)."

> *Referee Comment: Fig.2S : would be good that the CB be contextualised ie by presenting a larger map of Central Africa with countries borders ....*

**Author Response:** We have added a geopolitical map with Congo Basin boundaries as an additional panel in Figure S2.

*Referee Comment:* *Lastly I would have liked seeing in this paper a short « perspective » section on the further analyses these results call for and the study limits (unfortunately ET at the monthly and CB scale that does not allow documenting spatial variations in the drivers of ET that might be significant nor fine temporal - daily or infra-daily – variations which might be key to understand differences between the two wet or the two dry periods)*

**Author Response:** We added a "Section 4.5: Opportunities for further study" to the end of the Discussion outlining some further analyses that would be useful. It reads:

[revised manuscript text omitted]